# Contextual Online Decision Making with Infinite-Dimensional Functional Regression

Haichen Hu [1 2]  Rui Ai [3]  Stephen Bates [4]  David Simchi-Levi [2 3]

## Abstract

Contextual sequential decision-making is fundamental to machine learning, with applications in bandits, sequential hypothesis testing, and online risk control. These tasks often rely on statistical measures like expectation, variance, and quantiles. In this paper, we propose a universal algorithmic framework that learns the full underlying distribution, enabling a unified approach to all contextual online decision-making problems. The challenge lies in the uncountably infinite-dimensional regression, where existing contextual bandit algorithms all yield infinite regret. We innovatively propose an efficient infinite-dimensional functional regression oracle for contextual cumulative distribution functions (CDFs) and model every datum as a combination of context-dependent CDF basis functions. Our analysis reveals that the decay rate of the eigenvalue sequence of the design integral operator governs the regression error rate, and consequently, the utility regret rate. Specifically, when the eigenvalue sequence exhibits a polynomial decay of order $\frac{1}{\gamma} \geq 1$, the utility regret is bounded by $\widetilde{\mathcal{O}}\left(T^{\frac{3\gamma+2}{2(\gamma+2)}}\right)$. The case that $\gamma = 0$ can recover the existing optimal rate in contextual bandits literature with finite-dimensional regression and so as exponential decay. We also provide a numerical method to compute the eigenvalue sequence of integral operators, enabling the practical implementation of our framework.

## 1. Introduction

Contextual sequential online experimentation has been overwhelmingly important for online platforms, healthcare companies, and other businesses (Tewari & Murphy, 2017; Saha et al., 2020; Beygelzimer et al., 2011; Avadhanula et al., 2022). A decision maker (DM) might face a wide range of tasks, and different tasks lead to various adaptive algorithm designs. For example, for maximizing the total reward, there are all kinds of contextual bandit algorithms (Chu et al., 2011). For minimizing the total hypothesis testing error, we have online hypothesis testing algorithms (Wei et al., 2007). To rectify a pre-trained machine learning model, many online calibration algorithms have been developed (Fasiolo et al., 2021).

Some common structures appear to be obscured behind these examples. At each round, the DM first receives some context, such as past consumption records or symptoms in a healthcare setting. Then, the DM chooses an action to apply according to this context and his objective. For example, in online hypothesis testing, the actions could be "reject" and "accept", while in contextual bandits, the action is to choose and pull an arm. Commonly, there is an underlying distribution $P_{x,a}$ associated with every context $x$ and action $a$. The DM makes decisions based on the estimation of its various statistics. In the context of bandits, the primary statistic of interest is the expectation, whereas in risk control, attention is directed toward variance (Li et al., 2022; Füss et al., 2024). Please refer to Ayala-Romero et al. (2024); Bouneffouf (2016); Sun et al. (2016); Li et al. (2019); Han et al. (2021); Kong (2024) for more examples.

The essential difficulty behind all these online decision-making examples is the uncertainty of the outcomes under different actions. In revenue management, the decision maker is not sure about the mean reward, while in financial portfolio management, he is mostly uncertain about both mean and variance (Gupta et al., 2021). Moreover, in online quantile calibration, the key concern turns to the unknown quantile function (Bastani et al., 2022).

Note that no matter what kind of statistic we aim to know, as long as we learn the distribution $P_{x,a}$, the decision maker knows exactly how to make the decision. Therefore, it raises

[1]Center for Computational Science and Engineering, MIT, Cambridge, United States [2]Department of CEE, MIT, Cambridge, United States [3]Institute for Data, Systems, and Society, MIT, Cambridge, United States [4]Department of EECS, MIT, Cambridge, United States. Correspondence to: Haichen Hu <huhc@mit.edu>.

*Proceedings of the 42nd International Conference on Machine Learning*, Vancouver, Canada. PMLR 267, 2025. Copyright 2025 by the author(s).

a question:

*Is it possible to consider a more general (robust and adaptive) online decision-making setting based on infinite-dimensional functional regression?*

Specifically, we are looking for a general solution with infinite-dimensional functional regression of cumulative distribution functions (CDFs).

Another motivation for considering infinite-dimensional models is the practical success of high-dimensional models, such as AI products based on transformer architectures. Traditional finite-dimensional analysis fails to explain the strong performance of deep learning models. To fully address this challenge, our paper directly focuses on the infinite-dimensional regression problem.

**Roadmap:** We present our results in the following structure. In Section 2.1, we introduce our general sequential decision-making model with infinite-dimensional functional regression. Then in Section 2.2, we introduce the mathematical concepts, tools, and assumptions in functional analysis and operator theory that are essential for our infinite-dimensional regression. In Section 3, we put forward an efficient algorithm to carry out the functional regression and establish its oracle inequality. We further present our sequential decision-making procedure (Algorithm 2) in Section 4 and prove its theoretical guarantee in theorem 4.1.

## 1.1. Related Works

Our work is intimately related to the lines of work on contextual bandits (Tewari & Murphy, 2017; Zhou et al., 2023; Zhou, 2015; Bouneffouf et al., 2020; Agarwal et al., 2014; Chu et al., 2011), operation learning (Mollenhauer et al., 2022; Kovachki et al., 2024; Adcock et al., 2024; Foster et al., 2018; Foster & Rakhlin, 2020) and functional regression (Morris, 2015; Ramsay & Silverman, 2002; Zhang et al., 2022; Azizzadenesheli et al.; Yeh et al., 2023; Hou et al., 2023). Due to limited space, we discuss these and more related works in Appendix A.

**Notation:** For any measure space $(B, \mathcal{B}, n)$, we use $L^2(B, n)$ to denote the square-integrable function space which is also a Hilbert space with inner product $\langle \cdot, \cdot \rangle = \int_B f(x)g(x)dn(x)$ and norm $||f||_{L^2(B,n)} = \sqrt{\langle f, f \rangle}$. $\mathcal{O}(\cdot)$ and $o(\cdot)$ stand for Bachmann–Landau asymptotic notations up to constants. Meanwhile, $\widetilde{\mathcal{O}}$ stands for the asymptotic notations up to logarithmic terms. For any $n \in \mathbb{N}$, $[n]$ denotes the set $\{1, \cdots, n\}$, and $\mathbb{I}_y(t)$ denotes the indicator function $\mathbb{I}\{y \leq t\}$. Unless otherwise stated, when we write the eigenvalue sequence $\{\lambda_i\}$, it is arranged in a decreasing order.

## 2. Model setup and Operator Eigendecay

### 2.1. Model Setup

There are two main bodies of our framework, *Functional Regression Model* and *Decision-Making Model*. We first focus on the functional regression model.

**Infinite-dimensional Functional-Regression:** Assume a feature space $\mathcal{X}$ and a finite action space $\mathcal{A} = \{a_1, \cdots, a_K\}$. For any context $x \in \mathcal{X}$ and action $a$, there is an associated random variable $Y_{a,x}$ which takes values in some bounded Borel set $S \subset \mathbb{R}$ with $m(S) = 1$, where $m$ is some measure on the real line [♮]. We assume $Y_{a,x}$ has a cumulative distribution function $F^*(x, a, s)$ satisfying the following Assumption 2.1 and Assumption 2.2.

**Assumption 2.1.** We have access to a function family of CDF basis $\Phi = \{\phi(x, a, w, s)\}_{w \in \Omega}$ indexed by a compact set $\Omega \subset \mathbb{R}^d$ with $d$-dimensional Lebesgue measure $\nu(\Omega) = 1$. That is, given any $x \in \mathcal{X}, a \in \mathcal{A}, w \in \Omega$, $0 \leq \phi(x, a, w, s) \leq 1$ is a CDF of some $S$ valued random variable. For any $x, a$, there is an unknown non-negative coefficient function $\theta^* \in L^2(\Omega, \nu)$ such that

$$F^*(x, a, s) = \int_\Omega \theta^*(w)\phi(x, a, w, s)d\nu(w)$$
$$= \langle \theta^*(\cdot), \phi(x, a, \cdot, s) \rangle .$$

We also set $\int_\Omega \theta^*(w)d\nu(w) = 1$ to ensure that $F^*(x, a, s)$ is a cumulative distribution function. For boundedness, we assume that $||\theta^*||_{L^2(\Omega,\nu)} \leq M$ for some constant $M$.

The dimension of this functional regression problem is infinite, as the number of the functions in candidate model class $\Phi$ can be uncountably infinite, which makes solving this problem extremely hard. Modeling the basis model class $\Phi$ could be problem-driven; different tasks have different families. Generally, we use spline functions, trigonometric functions, truncated Gaussian mixtures, neural networks, and the Bernoulli random variable mixture to model the basis distribution family in different applications. Please see Zhang et al. (2022) for numerical details.

Furthermore, we assume that the candidate model class $\Phi$, parametrized by the set $\Omega$, has Lipschitz-continuity with respect to $w \in \Omega$. This assumption is common in learning theory (see Xu & Zeevi (2020)). Our goal in our functional regression model is to derive an accurate coefficient function estimation $\widehat{\theta}_\mathcal{D}$.

**Assumption 2.2.** $\phi(x, a, \cdot, s)$ is $L_0$-Lipschitz continuous, $|\phi(x, a, w, s) - \phi(x, a, r, s)| \leq L_0||w - r||_\infty$.

We now turn to our Decision-Making Model.

---

[♮] $m(S)$ can be any finite number. Our $m(S) = 1$ assumption here is just for notation convenience.

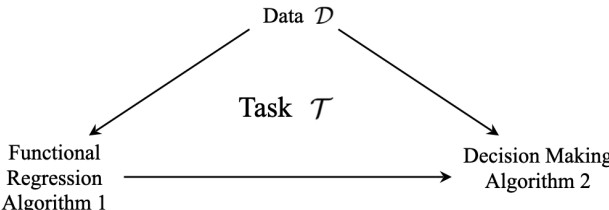

*Figure 1.* Data $\mathcal{D}$ drives the functional regression and decision-making for statistical task $\mathcal{T}$.

**Decision-making Model:** To incorporate all the sequential decision-making examples in our framework, we assume that the objective of the decision-maker is relevant to the cumulative distribution function itself, rather than only some concrete statistics such as mean, variance, quantile and so on.

To be specific, we assume that there is a known utility functional $\mathcal{T}$ defined on function space $L^2(S, m)$. At each round $t$, the context $x_t$ is drawn i.i.d. from some unknown distribution $Q_{\mathcal{X}}$. Given any context $x$, if the DM applies action $a$, then one data point will be sampled and observed according to distribution function $F^*(x, a, s)$, and the utility of action $a$ under $x$ is $\mathcal{T}(F^*(x, a, s))$. The goal of the decision maker is to maximize the total utility in $T$ rounds of interaction, i.e., $\max_{\{a_t\}_{t=1}^T} \sum_{t=1}^T \mathcal{T}(F^*(x_t, a_t, s))$.

Analogous to the bandit setting, the policy $\pi$ is a mapping from the feature space $\mathcal{X}$ to the action space probability simplex $\Delta(\mathcal{A})$. Given any $x$, we denote $a^*(x)$ as the action that maximizes the utility, i.e., $a^*(x) = \operatorname{argmax}_a \mathcal{T}(F^*(x, a, s))$. For any policy sequence $\{\pi_t\}_{t=1}^T$, the performance measure is utility regret, which is the difference between the optimal policy $\pi^*(x) = \delta_{a^*(x)}(\cdot)^\dagger$ and $\{\pi_s\}_{s=1}^T$:

$$\operatorname{Reg}(\{\pi_t\}_{t=1}^T)$$
$$\triangleq \sum_{t=1}^T \mathcal{T}(F^*(x_t, a^*(x_t), \cdot)) - \mathcal{T}(F^*(x, \pi_t(x), \cdot)).$$

**Assumption 2.3.** The known functional $\mathcal{T}$ is $L$-Lipschitz continuous with respect to the norm $||\cdot||_{L^2(S,m)}$.

Assumption 2.3 is essential since it ensures that an accurate estimation about the underlying distribution results in a good approximation of the corresponding utility functional value. Finally, we end this section with some practical applications and their corresponding utility functionals $\mathcal{T}$ and their Lipschitz constant $L$.

*Example* 2.4 (Contextual Bandits). Define the functional as $\mathcal{T}: F \to \int_S x dF(x)$, which is a known functional in terms of $F$. Given any context $x$, $F^*(x, a, s)$ is the conditional

---

$^\dagger \delta_a$ is the Dirac function of $a$, i.e., $\delta_a(x) = 1$ only if x=a, otherwise, $\delta_a(x) = 0$

reward distribution of arm $a$ given context $x$. Consequently, $\mathcal{T}(F^*(x, a, s))$ is the mean reward of arm $a$ given $x$, and we recover the contextual bandit problem.

We assume that the reward is bounded in interval $S = [a, b]$ and illustrate that contextual bandits satisfy our model assumption in Section 2. Note that $\mathcal{T}(F) = \int x dF(x)$, and integrating by parts, we have that for any two CDFs $F_1$, $F_2$,

$$|\mathcal{T}(F_1) - \mathcal{T}(F_2)| = \left| \int_S F_1(s) - F_2(s) ds \right|$$
$$\leq (b-a)^{1/2} ||F_1 - F_2||_{L^2(S)},$$

The inequality is by integrating by parts and Cauchy-Schwarz inequality. So contextual bandit satisfies Assumption 2.3 with constant $L = (b-a)^{1/2}$.

*Example* 2.5 (Sequential Hypothesis Testing (Naghshvar & Javidi, 2013)). $M$ hypotheses $\{H_1, \cdots, H_M\}$ are of interest, among which only one holds true. $a_i$ is to choose hypothesis $H_i$ to believe. For any true class $H_i$ and DM's judgment $a_j$, there is a loss function $l(H_i, a_j)$. The distribution associated with observation $x$ and $a_j$ is the posterior distribution of the loss $l$ with PMF denoted by $F^*_{x,a_j}$ $F^*_{x,a_j}(l(H_i, a_j)) = \mathbb{P}(H_i|x, a_j)$. $\mathbb{P}(H_i|x, a_j)$ is the posterior probability that $H_j$ is true given observation $x$ and $a_j$. $\mathcal{T}: F \to -\int_S s dF(s)$ is the negative expectation functional, where the integral stands for the Lebesgue integral. From the deduction of Example 2.4, we know that sequential hypothesis testing satisfies Assumption 2.3.

*Example* 2.6 (Mean-Variance (MV) bandits (Hu et al., 2022)). In multi-armed bandits (MAB), denote $F_i$ as the distribution of arm $i$. The mean-variance functional $\mathcal{T}: F_i \mapsto \rho \int x dF_i(x) - (1-\rho)(\int x^2 dF_i(x) - (\int x dF_i(x)^2))$. MV-MAB is about finding a policy that maximizes the total mean-variance. Similar to Example 2.4, we illustrate that this example satisfies Assumption 2.3.

$$|\mathcal{T}(F_1) - \mathcal{T}(F_2)|$$
$$\leq \rho \left| \int_S s dF_1(s) - \int_S s dF_2(s) \right|$$
$$+ (1-\rho)(b-a) \left| \int_S s dF_1(s) - \int_S s dF_2(s) \right|$$
$$\leq (\rho + (1-\rho)(b-a))(b-a)^{1/2} ||F_1 - F_2||_{L^2(S)}.$$

Therefore, mean-variance bandits satisfy Assumption 2.3 with Lipschitz constant $L = (\rho + (1-\rho)(b-a))(b-a)^{1/2}$.

It is intuitive that a good estimation of the underlying distribution may result in a good utility estimation. However, estimating the underlying CDF with an uncountable basis and arbitrary feature space is highly non-trivial. In Section 2.2, we would like to introduce some tools and assumptions from the functional analysis and operator theory community before we dive into our regression oracle. For

background knowledge of operator theory and functional analysis, please refer to some textbooks Conway (2000); Kubrusly (2011); Simon (2015).

## 2.2. Operator Eigendecay

Recall that in Section 2.1, we invoke the infinite-dimensional online decision-making problem, which makes the finite-dimensional linear algebra analysis invalid. Therefore, we turn to functional analysis and operator theory in infinite-dimensional spaces to handle our learning problem instead. One core object of the intersection regarding functional analysis and learning theory is the **operator**, which is intrinsically defined as mappings from infinite-dimensional spaces to infinite-dimensional spaces. In functional analysis, a very powerful tool to study the properties of many operators, especially Hilbert-Schmidt operators, is their eigenvalue and eigenfunction sequences. Specifically, in this paper, we care about the decay rate of the eigenvalue sequence (**eigendecay**), which is the essential factor that determines our approximation accuracy (theorem 3.6) and utility regret bound (theorem 4.1). In this subsection, we present all the mathematical concepts, theorems, and assumptions regarding the eigendecay rate of several important operators which will be utilized in Section 3.

**Definition 2.7.** For any $(w, r) \in \Omega \times \Omega$, define the following linear integral operator, integral kernel, and mapping:

$$\mathcal{L}^{x,a} : \theta(w) \mapsto \iint_{\Omega} \int_S \theta(r)\phi(x, a, w, s)\phi(x, a, r, s)dm(s)d\nu(r),$$

$$\mathcal{K}^{x,a} : (w, r) \mapsto \int_S \phi(x, a, w, s)\phi(x, a, r, s)dm(s),$$

$$\psi_{x,a} : L^2(\Omega, \nu) \times S \to \mathbb{R}, \ (\theta, s) \mapsto \langle \theta(\cdot), \phi(x, a, \cdot, s) \rangle.$$

By Fubini's theorem, we have

$$\forall w \in \Omega, \ \mathcal{L}^{x,a}(\theta)(w) = \int_{\Omega} \theta(r)\mathcal{K}^{x,a}(w, r)d\nu(r)$$
$$= \int_S \phi(x, a, w, s)\psi_{x,a}(\theta, t)dm(s).$$

**Theorem 2.8.** *For any $x \in \mathcal{X}$ and $a \in \mathcal{A}$, $\mathcal{L}^{x,a}$ is a compact, linear, positive and self-adjoint Hilbert-Schmidt integral operator with $\|\mathcal{L}^{x,a}\| \leq 1$.*

By the Riesz-Schauder theorem and the spectral decomposition theorem (Reed & Simon, 1978), if we denote $\{\lambda_i(\mathcal{L})^{x,a}\}_{i=1}^{\infty}$ as its eigenvalue sequence in decreasing order and $\{e_i(\mathcal{L}^{x,a})\}$ as the corresponding eigenfunctions, then it holds that

$$\mathcal{L}^{x,a}(\theta) = \sum_{i=1}^{\infty} \lambda_i(\mathcal{L}^{x,a}) \langle \theta, e_i(\mathcal{L}^{x,a}) \rangle e_i(\mathcal{L}^{x,a}),$$

where $\lambda_1(\mathcal{L}^{x,a}) \geq \lambda_2(\mathcal{L}^{x,a}) \geq \cdots > 0$ and $\lambda_i(\mathcal{L}^{x,a}) \to 0$. The eigenvalues and eigenfunctions play a crucial role in our algorithm and analysis. Hence, we further examine them and state additional properties below. All proofs are deferred to Appendix C.

By Gohberg et al. (2012); Ferreira & Menegatto (2013), we call the Hilbert-Schmidt integral operator $\mathcal{L}^{x,a}$ traceable if $\sum_{i=1}^{\infty} \lambda_i(\mathcal{L}^{x,a}) < \infty$. For any traceable operator, we could analyze the decay rate of its eigenvalue sequence, and the eigenvalue decay rate of these integral operators has been well studied in functional analysis (Ferreira & Menegatto, 2013; Volkov, 2024; Levine, 2023; Carrijo & Jordão, 2020).

Specifically, for our integral operator $\mathcal{L}^{x,a}$, we observe the following properties.

*Property* 2.9. For any $x, a$, the integral operator $\mathcal{L}^{x,a}$ satisfies

- $\sum_{i=1}^{\infty} \lambda_i(\mathcal{L}^{x,a}) = \int_{\Omega} \int_S \phi(x, a, w, s)^2 dm(s)d\nu(w) \leq 1$.

- For any $i$, $\exists C > 0$ such that $\lambda_i(\mathcal{L}^{x,a}) \leq \frac{C}{i}$.

- $\lambda_i(\mathcal{L}^{x,a}) = o(\frac{1}{i})$.

For many concrete integral operators, Property 2.9 could be further strengthened and the decay rate of the eigenvalue sequence could be established. For example, one might show polynomial eigendecay and even exponential eigendecay (Carrijo & Jordão, 2020). Intuitively, a larger eigenvalue implies that more 'information' is stored in the direction of the corresponding eigenfunction. Therefore, faster decaying of the eigenvalue sequence means that more 'information' is concentrated in several large eigenvalues and the corresponding eigenfunctions. In this paper, we use a single positive parameter $\gamma$ in Assumption 2.10 to describe the decay rate of the eigenvalue sequence and characterize this phenomenon, which is called $\gamma$-**dominating eigendecay**.

**Assumption 2.10** ($\gamma$-dominating eigendecay)**.** Denoting all the bounded linear operators on $L^2(\Omega, \nu)$ as $\mathcal{B}(L^2(\Omega, \nu))$, we assume that the set $\{\mathcal{L}^{x,a} : x \in \mathcal{X}, a \in \mathcal{A}\} \subset \mathcal{B}(L^2(\Omega, \nu))$ is convex and satisfies the existence of a sequence $\{\tau_i\}_{i=1}^{\infty}$ such that

- for some $0 < \gamma \leq 1$ and $s_0 < \infty$, $\sum_{k=1}^{\infty} \tau_k^{\gamma} \leq s_0$,

- for $\forall x, a, \ \lambda_k(\mathcal{L}^{x,a}) < \tau_k \leq \mathcal{O}(\frac{1}{k}), \ \forall k$.

We name the sequence $\{\tau_i\}_{i=1}^{\infty}$ as "$\gamma$-dominating sequence". Intuitively, Assumption 2.10 states that the decay rate of the eigenvalue sequence of any operator $\mathcal{L}$ in the integral operator set $\{\mathcal{L}^{x,a} : x \in \mathcal{X}, a \in \mathcal{A}\}$ could be "dominated" by some polynomially converging series. The rate parameter $\gamma$ is a key factor in our analysis. In Section 3, we will prove that parameter $\gamma$ influences the error rate of the functional

regression oracle and therefore determines our utility regret rate.

Finally we remark that our assumption in Assumption 2.10 essentially claims that the eigenvalue sequence $\{\lambda_k\}$ decays with a polynomial order in magnitude. It is quite common to assume polynomial or even exponential eigendecay rate of large matrix or neural network operators (Yeh et al., 2023; Vakili & Olkhovskaya, 2024; Goel & Klivans, 2017; Agarwal & Gonen, 2018).

# 3. Functional Regression and Oracle Inequality

## 3.1. Functional Regression

In Section 3.1, we provide a method about how to estimate our coefficient function $\theta^*(w)$, given a dataset $\mathcal{D} = \{(x_j, a_j, y_j)\}_{j=1}^n$. Here, $y_j$ is sampled according to CDF $F^*(x_j, a_j, s)$.

The intuition of functional regression oracle is to use the operator's spectral decomposition. In ordinary least squares regression $\min \|\boldsymbol{Y} - \boldsymbol{X}\theta\|^2$, the normal matrix $\boldsymbol{X}^T \boldsymbol{X}$ plays a crucial role (Goldberger, 1991). Here, we require an operator that behaves similarly.

**Definition 3.1** (Design Integral Operator). For any given dataset $\mathcal{D} = \{(x_j, a_j, y_j)\}_{j=1}^n$ whose $|\mathcal{D}| = n$, we define the design integral operator $\mathcal{U}_{\mathcal{D}}$ as $\forall w \in \Omega$,

$$
\begin{aligned}
\mathcal{U}_{\mathcal{D}}(\theta)(w) &\triangleq \sum_{j=1}^n \mathcal{L}^{x_i, a_i}(\theta)(w) \\
&= \sum_{j=1}^n \int_{\Omega} \theta(r) \mathcal{K}^{x_j, a_j}(w, r) d\nu(r).
\end{aligned}
$$

Moreover, we denote the spectral decomposition of $\mathcal{U}_{\mathcal{D}}$ as $\sum_{i=1}^{\infty} \lambda_i(\mathcal{U}_{\mathcal{D}}) e_{\mathcal{U}_{\mathcal{D}}}^i$.

Using Theorem 2.8, one could verify that $\mathcal{U}_{\mathcal{D}}$ is also a linear, positive, self-adjoint and Hilbert-Schmidt integral operator. Therefore, we have $\lambda_i(\mathcal{U}_{\mathcal{D}}) \to 0$, and $\mathcal{U}_{\mathcal{D}}$ is not invertible.

In finite-dimensional regression, people often handle non-invertibility with regularization terms $\lambda\|\theta\|^2$, leading to an error that scales with the problem dimension. However, since our dimension is infinite, adding this regularization term is infeasible. Zhang et al. (2022) considers a different data-driven regularization, but the term will scale with the cardinality $|\mathcal{D}| = n$, thus it's inapplicable to obtain convergence results.

Rather than regularization, we consider a truncation of the function series $\sum_{i=1}^{\infty} \lambda_i(\mathcal{U}_{\mathcal{D}}) e_{\mathcal{U}_{\mathcal{D}}}^i$ according to the eigenvalue sequence $\{\lambda_i(\mathcal{U}_{\mathcal{D}})\}$. Specifically, we define the truncated finite rank operator $\widehat{\mathcal{U}}_{\mathcal{D}, \varepsilon}$.

**Definition 3.2.** For any small number $\varepsilon > 0$, the truncated integral operator $\widehat{\mathcal{U}}_{\mathcal{D}, \varepsilon}$ is defined as

$$
\widehat{\mathcal{U}}_{\mathcal{D}, \varepsilon} : \theta \mapsto \sum_{i=1}^{N_{\mathcal{D}, \varepsilon}} \lambda_i(\mathcal{U}_{\mathcal{D}}) \langle \theta, e_{\mathcal{U}_{\mathcal{D}}}^i \rangle e_{\mathcal{U}_{\mathcal{D}}}^i,
$$

where $N_{\mathcal{D}, \varepsilon}$ is the smallest number such that for $\forall i \geq N_{\mathcal{D}, \varepsilon} + 1$, $\lambda_i(\mathcal{U}_{\mathcal{D}}) < n\varepsilon$.

$\varepsilon$ is some hyper-parameter which will be determined later. Since we arrange the eigenvalues in a decreasing order, we have that for $\forall i \leq N$, $\lambda_i(\mathcal{U}_{\mathcal{D}}) \geq n\varepsilon$. Despite the fact that $\widehat{\mathcal{U}}_{\mathcal{D}, \varepsilon}$ is not invertible, it is still possible to define its pseudo-inverse as follows.

**Definition 3.3.** The pseudo-inverse of the operator $\widehat{\mathcal{U}}_{\mathcal{D}, \varepsilon}$ is defined as

$$
\widehat{\mathcal{U}}_{\mathcal{D}, \varepsilon}^{\dagger} : \theta \mapsto \sum_{i=1}^{N_{\mathcal{D}, \varepsilon}} \frac{1}{\lambda_i(\mathcal{U}_{\mathcal{D}})} \langle \theta, e_{\mathcal{U}_{\mathcal{D}}}^i \rangle e_{\mathcal{U}_{\mathcal{D}}}^i.
$$

We set $\varepsilon = \varepsilon^* \triangleq n^{-\frac{2}{\gamma+2}}$ to conduct functional regression. With a bit of abuse of notation, we also use $N_{\varepsilon}$ to denote $N_{\mathcal{D}, \varepsilon}$, and abbreviate $\widehat{\mathcal{U}}_{\mathcal{D}, \varepsilon^*}, \widehat{\mathcal{U}}_{\mathcal{D}, \varepsilon^*}^{\dagger}$ as $\widehat{\mathcal{U}}_{\mathcal{D}}$ and $\widehat{\mathcal{U}}_{\mathcal{D}}^{\dagger}$.

After obtaining $\widehat{\mathcal{U}}_{\mathcal{D}}$ and $\widehat{\mathcal{U}}_{\mathcal{D}}^{\dagger}$, there are two remaining steps of our functional regression. The first step is to compute the following function:

$$
\theta_{\mathcal{D}} = \mathcal{U}_{\mathcal{D}}^{\dagger} \left( \int_S \sum_{j=1}^n \mathbb{I}_{y_j}(t) \phi(x_j, a_j, w, s) dm(s) \right).
$$

The motivation for calculating it lies in its ability to solve the following least squares optimization problem presented in Theorem 3.4. Intuitively, for any data point $(x_i, a_i)$, the underlying targeted distribution function is $F^*(x_i, a_i, s) = \int_{\Omega} \theta^*(w) \phi(x_i, a_i, w, s) d\nu(w)$. Normally, to learn the function $\theta^*(w)$, we need to observe the whole function $F^*(x_i, a_i, s) = \int_{\Omega} \theta^*(w) \phi(x_i, a_i, w, s) d\nu(w)$ and solve an inverse problem to obtain $\theta^*$. However, due to our bandit feedback nature, we only have one sample $y_i$ from this distribution. Thus, considering the $L^2$ error between the real empirical counterpart $\mathbb{I}_{y_i}(t)$ and any candidate distribution function $\int_{\Omega} \theta(w) \phi(x_i, a_i, w, s) d\nu(w)$ for all $n$ data points, we obtain the following loss function $l(\theta, \mathcal{D})$ in Theorem 3.4.

**Theorem 3.4.** We define the loss function $l(\theta, \mathcal{D})$ on dataset $\mathcal{D}$ as

$$
l(\theta, \mathcal{D}) \triangleq \sum_{j=1}^n \|\mathbb{I}_{y_i}(s) - \int_{\Omega} \theta(w) \phi(x_i, a_i, w, s) d\nu(w)\|_{L^2(S, m)}^2.
$$

Then, $\theta_{\mathcal{D}}$ solves the following optimization problem,

$$
\boldsymbol{P}) : \min_{\theta \in \mathrm{Span}(e_1, \cdots, e_{N_{\varepsilon^*}})} l(\theta, \mathcal{D}), \tag{1}
$$

*where* $\mathrm{Span}(X)$ *denotes the linear subspace spanned by the vector group* $X$.

After obtaining $\theta_{\mathcal{D}}$, in order to ensure that our estimated coefficient function satisfies Assumption 2.1, we project it onto the set $\mathcal{C} = \left\{ \theta \geq 0 : \int_{\Omega} \theta d\nu = 1, ||\theta||_{L^2(\Omega,\nu)} \leq M \right\}$ in the following norm.

**Definition 3.5.** We define the following norm for any positive self-adjoint bounded linear operator $\mathcal{U}$:

$$||\theta||_{\mathcal{U}} \triangleq \sqrt{\langle \theta, \mathcal{U}(\theta) \rangle}.$$

Correspondingly, we define the projection $\mathcal{P}_{\mathcal{D}}$ onto set $\mathcal{C}$ under the norm $|| \cdot ||_{\mathcal{U}_{\mathcal{D}}}$ as

$$\mathcal{P}_{\mathcal{D}}(x) \triangleq \underset{y \in \mathcal{C}}{\mathrm{argmin}} ||y - x||_{\mathcal{U}_{\mathcal{D}}}^2.$$

The second step is to project $\theta_{\mathcal{D}}$ onto set $\mathcal{C}$ by $\mathcal{P}_{\mathcal{D}}$, i.e., $\widehat{\theta}_{\mathcal{D}} \triangleq \mathcal{P}_{\mathcal{D}}(\theta_{\mathcal{D}})$. This process yields our estimated coefficient function $\widehat{\theta}_{\mathcal{D}}$. At the end of this section, we summarize the functional regression oracle FuncReg in Algorithm 1.

---

**Algorithm 1** Regression Oracle: FuncReg

---

**Require:** Basis function family $\{\phi(x, a, w, t)\}_{w \in \Omega}$, dataset $\mathcal{D} = \{x_i, a_i, y_i\}_{i=1}^n$, $\{\tau_i\}_{i=1}^\infty$, $0 < \gamma \leq 1$.
Define the integral operator

$$\mathcal{U}_{\mathcal{D}}(\theta)(w) = \sum_{j=1}^{n} \mathcal{L}^{x_i, a_i}(\theta)(w)$$

$$= \sum_{j=1}^{n} \int_{\Omega} \mathcal{K}^{x_j, a_j}(w, r)\theta(r)d\nu(r).$$

Compute the spectral decomposition of $\mathcal{U}_{\mathcal{D}}{}^*$, say

$$\mathcal{U}_{\mathcal{D}}(\theta) = \sum_{i=1}^{\infty} \lambda_i \langle \theta, e_i \rangle e_i.$$

Define $\varepsilon^* = n^{-\frac{2}{\gamma+2}}$ and construct the truncated operator $\widehat{\mathcal{U}}_{\mathcal{D}}$ and its pseudo-inverse $\widehat{\mathcal{U}}_{\mathcal{D}}^\dagger$ according to Definition 3.2 and Definition 3.3.
Solve the optimization problem 1 and obtain solution $\theta_{\mathcal{D}}$.
Compute the projection $\widehat{\theta}_{\mathcal{D}} = \mathcal{P}_{\mathcal{D}}(\theta_{\mathcal{D}})$.

**Return:** $\widehat{\theta}_{\mathcal{D}}$.

---

### 3.2. Oracle Inequality

In this section, we give two versions of oracle inequalities. The first one is for the fixed design that $\mathcal{D}$ is arbitrarily

---

given, whereas the second one is for the random design that $\{(x_i, a_i)\}_{i=1}^n$ are sampled from some unknown joint distribution $Q$. ,

*We present a concrete numerical method to compute eigenvalues in Appendix F.

**Theorem 3.6.** *Under Assumption 2.1 and Assumption 2.10, given dataset* $\mathcal{D} = \{(x_j, a_j, y_j)\}_{j=1}^n$ *and setting* $\varepsilon^* = n^{-\frac{2}{\gamma+2}}$, *then with probability at least* $1 - \delta$, *the estimated coefficient function* $\widehat{\theta}_{\mathcal{D}}$ *from Algorithm 1 satisfies*

$$||\widehat{\theta}_{\mathcal{D}} - \theta^*||_{\mathcal{U}_{\mathcal{D}}} \leq \mathcal{E}_{\mathcal{D}}^\delta(n)$$

$$\triangleq \sqrt{2 \left( 2\log(\frac{1}{\delta}) + \sum_{i=1}^{N_{\varepsilon^*}} \log(1 + \lambda_i(\mathcal{U}_{\mathcal{D}})) \right)} + n^{\frac{\gamma}{\gamma+2}} M.$$

Theorem 3.6 provides a theoretical guarantee about our regression error when the given $\mathcal{D}$ has cardinality $|\mathcal{D}| = n$. Nevertheless, Theorem 3.6 cannot be directly used, as there exists an agnostic term $\sum_{i=1}^{N_{\varepsilon^*}} \log(1 + \lambda_i(\mathcal{U}_{\mathcal{D}}))$ related to the dataset $\mathcal{D}$. The following important lemma gives us an upper bound of $\mathcal{E}_{\mathcal{D}}^\delta(n)$ which is independent of $\mathcal{D}$.

**Lemma 3.7.** *For any dataset* $\mathcal{D} = \{(x_i, a_i, y_i)\}_{i=1}^n$ *whose* $|\mathcal{D}| = n$, *by choosing* $\varepsilon = \varepsilon^* = n^{\frac{-2}{\gamma+2}}$, *we have*

$$\mathcal{E}_{\mathcal{D}}^\delta(n) = \sqrt{2 \left( 2\log(\frac{1}{\delta}) + \sum_{i=1}^{N_{\varepsilon^*}} \log(1 + \lambda_i) \right)} + n^{\frac{\gamma}{\gamma+2}} M$$

$$\leq \mathcal{E}_\delta(n) \triangleq 2\log(1/\delta)^{1/2} + \left( 2\sqrt{s_0 \log(1 + n)} + M \right) n^{\frac{\gamma}{\gamma+2}}.$$

With Lemma 3.7, we derive the following corollary which is independent of $\mathcal{D}$.

**Corollary 3.8.** *Under Assumption 2.1 and Assumption 2.10, given dataset* $\mathcal{D} = \{(x_j, a_j, y_j)\}_{j=1}^n$ *and setting* $\varepsilon^* = n^{-\frac{2}{\gamma+2}}$, *then with probability at least* $1 - \delta$, *we have* $||\widehat{\theta}_{\mathcal{D}} - \theta^*||_{\mathcal{U}_{\mathcal{D}}} \leq \mathcal{E}_\delta(n)$.

When the data points are sampled i.i.d. according to some distribution, we could further give an oracle inequality for random design. Taking it into consideration, our oracle inequality for random design helps to bound the $L^2$ error between the ground truth CDF and our estimated CDF.

We first state some mild assumptions in random design.

**Assumption 3.9.** We assume that for any $x, a, w, r$, there exists a constant $\eta > 0$ such that $\mathcal{K}^{x,a}(w, r) = \int_S \phi(x, a, w, s)\phi(x, a, r, s)dm(s) \geq \eta > 0$.

Assumption 3.9 plays a role in guaranteeing the integral operator will not map a function far from $0 \in L^2(\Omega, \nu)$ to some near-zero point, which is a bit similar to the concept of proper mapping in convex analysis (Magaril-Il'yaev & Tikhomirov, 2003). With Assumption 2.1, we know that $\Omega \times \Omega$ is also compact in $\mathbb{R}^{2d}$. Thus, we have $\mathcal{N}(t, \Omega \times \Omega, ||\cdot||_\infty) \leq \left( \frac{A}{t} \right)^{2d}$ for some number $A$, delayed to Lemma B.9.

We finally transfer the estimation error of $\widehat{\theta}_{\mathcal{D}}$ to the $L^2$ error between cumulative distribution functions,

$$\|\widehat{\theta}_{\mathcal{D}} - \theta^*\|^2_{\mathcal{U}_{\mathcal{D}}}$$

$$= \left\langle \widehat{\theta}_{\mathcal{D}} - \theta^*, \mathcal{U}_{\mathcal{D}} \left( \widehat{\theta}_{\mathcal{D}} - \theta^* \right) \right\rangle$$

$$= \sum_{j=1}^{n} \left\langle \widehat{\theta}_{\mathcal{D}} - \theta^*, \mathcal{L}^{x_j, a_j} \left( \widehat{\theta}_{\mathcal{D}} - \theta^* \right) \right\rangle$$

$$= \sum_{j=1}^{n} \int_{\mathcal{X}, \mathcal{A}} \int_{S} \left( \int_{\Omega} \left( \widehat{\theta}_{\mathcal{D}} - \theta^* \right)(w) \phi(x_j, a_j, w, s) d\nu(w) \right)$$

$$\times \left( \int_{\Omega} \left( \widehat{\theta}_{\mathcal{D}} - \theta^* \right)(r) \phi(x_j, a_j, r, s) d\nu(r) \right) dm(s)$$

$$= \sum_{j=1}^{n} \left[ \|\widehat{F}_{\mathcal{D}}(x_j, a_j, s) - F^*(x_j, a_j, s)\|^2_{L^2(S, m)} \right].$$

Consequently, combining Theorem 3.6 with some concentration analysis and Lemma 3.7, we have the following theorem.

**Theorem 3.10.** *Given some dataset* $\mathcal{D} = \{(x_j, a_j, y_j)\}_{j=1}^{n}$ *where* $(x_j, y_j) \overset{i.i.d.}{\sim} Q$, *we define* $\widehat{F}_{\mathcal{D}}(x, a, s) \triangleq \int_{\Omega} \widehat{\theta}_{\mathcal{D}}(w) \phi(x, a, w, s) d\nu(w)$ *to be our estimated cumulative distribution function under* $x, a$. *For any* $0 \leq \delta \leq 1/2$, *we have that with probability at least* $1 - 2\delta$,

$$\mathbb{E}_{(x,a)\sim Q} \left[ \|\widehat{F}_{\mathcal{D}}(x, a, s) - F^*(x, a, s)\|^2_{L^2(S,m)} \right]$$

$$\leq \frac{C(d, L_0, \delta, A, \eta) \mathcal{E}_{\delta}(n)^2}{n},$$

*where*

$$C(d, L_0, \delta, A, \eta)$$

$$= \left( 1 + (48\sqrt{d \log(2L_0 A)} + 2\sqrt{\log(1/\delta)})/\eta \right)$$

*is some constant that does not influence the* $L^2$ *error rate with respect to* $n$.

# 4. Algorithm

We provide an algorithm in Section 4 that incorporates our functional regression oracle to minimize the expected utility regret.

It is computationally expensive to calculate eigenvalues and corresponding eigenfunctions of an integral operator. Hence, it is desirable to develop algorithms with a low-frequency oracle call property. Inspired by Simchi-Levi & Xu (2021); Qian et al. (2024), we design an Inverse Gap Weighting policy in a batched version possessing the low-frequency oracle call property.

Before delving into the specific algorithm, the following statement indicates that a well-bounded estimation error implies a well-bounded decision-making error.

**Theorem 4.1.** *Given some dataset* $\mathcal{D} = \{(x_j, a_j, y_j)\}_{j=1}^{n}$ *where* $(x_j, a_j, y_j) \overset{i.i.d.}{\sim} Q$, *we define* $\widehat{F}_{\mathcal{D}}(x, a, s) \triangleq \int_{\Omega} \widehat{\theta}_{\mathcal{D}}(w) \phi(x, a, w, s) d\nu(w)$ *as our estimated cumulative distribution function. For any* $0 < \delta < 1$, *we have that with probability at least* $1 - \delta$,

$$\mathbb{E}_{(x,a)\sim Q} \left[ \left( \mathcal{T}(\widehat{F}_{\mathcal{D}}(x, a, t)) - \mathcal{T}(F^*(x, a, t)) \right)^2 \right]$$

$$\leq \frac{L^2 C(d, L_0, \delta/2, A, \eta) \mathcal{E}_{\delta/2}(n)^2}{n},$$

*where*

$$C(d, L_0, \delta/2, A, \eta)$$

$$= \left( 1 + (48\sqrt{d \log(2L_0 A)} + 2\sqrt{\log(2/\delta)})/\eta \right).$$

We use $\mathsf{Est}_{\delta}(n)$ to denote $L^2 C(d, L_0, \delta, A, \eta) \mathcal{E}_{\delta}(n)^2$ thereafter. For any $x, a$, we could view $\mathcal{T}(F^*(x, a, s))$ as the unknown expected "reward" related to action $a$ given context $x$. Therefore, we transform our abstract sequential decision-making problem into a stochastic contextual bandit problem. Although in our problem, at every round, we do not directly observe a sample point from the "reward" distribution, we could still estimate it by functional regression FuncReg, yielding the desired reduction from the sequential decision-making problem to a contextual bandit.

We summarize our algorithm in Algorithm 2. For the algorithm structure, we first divide the whole $T$ rounds into several epochs and geometrically increase the length of every epoch so that the low-frequency oracle call property is automatically satisfied in Algorithm 2. At the beginning of every epoch $m$, we call our functional regression oracle FuncReg based on the i.i.d. data gathered from the last epoch to get an estimation $\widehat{\theta}_m$. We then design our inverse gap weighting policy $\pi_m$ based on $\widehat{\theta}_m$ and execute it throughout this epoch. By such a structure, we could ensure that the data generated throughout this epoch are i.i.d. so that we can use Theorem 3.10 to bound the $L^2$ estimation error, and therefore the utility regret.

To be specific, we first impose epoch schedule $\tau_m = 2^m$, which means that the $(m+1)$-th epoch is twice as long as the $m$-th epoch. Therefore, the statistical guarantee we get from epoch $m+1$ is stronger than that of epoch $m$. As $m$ scales, our estimation becomes more and more accurate. The Inverse Gap Weighting policy enables us to balance the exploration and exploitation trade-off to maintain a low regret just assuming access to an offline regression oracle with i.i.d. input data. We now provide the theoretical guarantee of our Algorithm 2 as follows.

**Theorem 4.2.** *For stochastic context setting, assuming that we can only call functional regression oracle* $\lceil \log(T) \rceil$

**Algorithm 2** Stochastic Contextual Decision Making with Infinite Functional Regression

---

**Require:** Functional $\mathcal{T}$, feature space $\mathcal{X}$, action space $\mathcal{A} = \{a_1 \cdots, a_K\}$ , basis function family $\{\phi(x, a, w, u)\}_{w \in \Omega}$, regression space $L^2(\Omega, \nu)$, range space of random variables $(S, m)$.

Initialize epoch schedule $0 = \xi_0 < \xi_1 < \xi_2 < \cdots$

**for** $m = 1, 2, \cdots$ **do**

Obtain $\widehat{\theta}_m \in L^2(\Omega, \nu)$ from oracle FuncReg with dataset $\mathcal{D}_{m-1} = \{(x_t, a_t, y_t)\}_{t=\tau_{m-2}+1}^{\tau_{m-1}}$,

$$\widehat{\theta}_m = \mathsf{FuncReg}(\mathcal{D}_{m-1}).$$

Define exploration parameter $\varsigma_m = \frac{1}{2}\sqrt{\frac{K}{\mathsf{Est}_{\delta/2m^2}(\tau_{m-1}-\tau_{m-2})}}$ (for epoch 1, $\varsigma_1 = 1$).

**for** Round $t = \xi_{m-1} + 1 \cdots, \xi_m$ **do**

Observe $x_t$.

Compute the value of functional for every action

$$\widehat{v}_m(a) = \mathcal{T}(\int_\Omega \widehat{\theta}_m \phi(x_t, a, w, s) d\nu(w)) \,\forall a \in \mathcal{A}.$$

Define $\mathsf{IGW}_{\varsigma_m}$ policy:

$$p_t(a) = \begin{cases} \frac{1}{K+\varsigma_m(\widehat{v}_m(\widehat{a}_m)-\widehat{v}_m(a))} & \text{for all } a \neq \widehat{a}_m \\ 1 - \sum_{a \neq \widehat{a}_t} p_t(a) & \text{for } a = \widehat{a}_m. \end{cases}$$

Sample $a_t \sim p_t$ and observe one data point from cumulative distribution function $F^*(x_t, a_t, s)$.

**end for**

**end for**

---

*times, then with probability at least $1 - \delta$, the expected regret of Algorithm 2 after $T$ rounds is at most*

$$\mathbb{E}[Reg(T)] \leq \widetilde{C}(K, L, L_0, A, d, \eta)\left((s_0^{1/2} + M) \vee 1\right) *$$
$$\mathcal{O}\left(\log(\log T/\delta)\left(\sqrt{T} + T^{\frac{3\gamma+2}{2(\gamma+2)}}\right)\right),$$

*where $\widetilde{C}(A, K, L, L_0, d, \eta)$ is some constant that is only relevant to the parameters in the bracket.*

As $0 < \gamma \leq 1$, the term $\sqrt{T}$ is dominated by $T^{\frac{3\gamma+2}{2(\gamma+2)}}$ and our regret rate becomes

$$\mathcal{O}\left(\widetilde{C}\left((s_0^{1/2} + M) \vee 1\right) T^{\frac{3\gamma+2}{2(\gamma+2)}}(\log(\log T/\delta))\right),$$

where $\widetilde{C} = \widetilde{C}(K, L, L_0, A, d, \eta)$.

From Theorem 4.2, we observe that the regret rate is determined by the decay speed of the eigenvalue sequence and we present a numerical method to compute these sequences in Appendix F. Specifically, when $\gamma \searrow 0$, our regret rate

is closer and closer to the $\sqrt{T}$ order, which matches the traditional optimal regret rate. If the eigenvalue sequence is decaying exponentially fast, then it is also decaying polynomially fast for any positive $\gamma > 0$. Thus, by choosing $\gamma \searrow 0$, we conclude that the regret rate has order $\widetilde{\mathcal{O}}\left(\sqrt{T}\right)$ for any exponentially decaying eigenvalue sequence. Moreover, for finite-dimensional problems, we could imagine that all the eigenvalues with an index larger than the dimension number are zero. So, we can also set $\gamma = 0$ in Theorem 4.2 and recover the optimal regret rate $\widetilde{\mathcal{O}}(T^{1/2})$ up to some logarithmic factors.

On the other hand, if we have no prior knowledge about the order of the dominating sequence $\{\tau_i\}_{i=1}^\infty$ but just know that it converges, we can simply set it to 1 and still achieve sublinear regret $\widetilde{\mathcal{O}}(T^{5/6})$, as shown in Corollary 4.3.

**Corollary 4.3.** *For the dominating sequence $\{\tau_i\}_{i=1}^\infty$, if we only have information that it converges without any knowledge of the order $\gamma$, we can set $\gamma = 1$ and get $\frac{3\gamma+2}{2(\gamma+2)} = \frac{5}{6}$. Therefore, we could obtain the following expected regret bound:*

$$\mathbb{E}[Reg(T)] \leq \widetilde{C}(A, K, L, L_0, d, \eta)\left((s_0^{1/2} + M) \vee 1\right) *$$
$$\mathcal{O}\left(T^{\frac{5}{6}}\log(\log T/\delta)\right).$$

In summary, our algorithm is adaptive and robust. It not only recovers the optimal regret rate in finite-dimensional problems and infinite-dimensional problems with exponential eigendecay but can still manage to achieve a sublinear regret with no prior information about the eigendecay rate as well. These properties demonstrate the versatility of our algorithm and its broad potential for application.

We finish this section with a remark that in the design of Algorithm 2, we do not involve any information about the total round number $T$. If we know $T$ in advance, we could further have a more efficient epoch schedule that reduces the offline functional regression oracle call times from $\mathcal{O}(\log T)$ to $\mathcal{O}(\log \log T)$ (Simchi-Levi & Xu, 2021; Qian et al., 2024).

## 5. Conclusion and Discussion

In this paper, we establish a general framework for stochastic contextual online decision-making with infinite-dimensional functional regression, which incorporates any application examples with Lipschitz continuous objective functionals. We study the relationship between the utility regret and the eigenvalue sequence decay of our desired integral operator. Compared with finite-dimensional linear bandits, this connection is new and crucial in functional regression with infinite dimensions. Furthermore, we design a computationally efficient algorithm to solve our sequential decision-making problem based on a novel infinite-dimensional functional regression oracle.

Finally, we would like to discuss some interesting future research directions and we leave these interesting questions as potential next steps.

**Extension to adversarial contextual decision-making:** The epoch of Algorithm 2 is designed for lower computation cost, through the low-frequency algorithm call property, of computing eigenvalues and eigenfunctions in our functional regression oracle FuncReg. We observe that the guarantee provided by our oracle in Theorem 3.6 inherently adapts to any dataset where the contexts are arbitrarily drawn. This raises a natural question: can we design an efficient online functional regression oracle (Foster & Rakhlin, 2020), built on Algorithm 1, to address adversarial contextual decision-making problems?

**Minimax lower bound:** In our paper, the utility regret rate is $\widetilde{\mathcal{O}}(T^{\frac{3\gamma+2}{2(\gamma+2)}})$. An interesting open question is whether this dependency on $\gamma$ is optimal. It is worth future exploration into the minimax lower bound of the utility regret with respect to the eigenvalue decay rate $\gamma$.

**Extension to nonlinear functional regression:** In our problem, the relationship between $F^*(x, a, s)$ and $\Phi = \{\phi(x, a, w, s)\}_{w \in \Omega}$ is linear because of the linearity of integration. One challenging problem is to extend our methodology to some potential nonlinearity between $F^*(x, a, s)$ and our basis function family and design efficient decision-making algorithms.

## Impact Statement

This paper presents work whose goal is to advance the field of Machine Learning. There are many potential societal consequences of our work, none which we feel must be specifically highlighted here.

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

## A. Related Works

**Contextual Bandits** Contextual bandits have been widely studied in both academia and industry, including applications such as recommendation systems (Tewari & Murphy, 2017) and healthcare (Zhou et al., 2023). Please refer to Zhou (2015); Bouneffouf et al. (2020) for comprehensive surveys. However, most theoretical results require assumptions about the dimensionality, such as the context having a finite cardinality (Agarwal et al., 2014) or the existence of a low-dimensional representation, as in linear contextual bandits (Chu et al., 2011). Our paper is the first to address the problem of contextual decision-making based on infinite-dimensional functional regression. We establish universal regret bounds for various tasks under eigenvalue decay assumptions and further derive general sublinear regrets without them.

**Operator Learning.** Operator learning (Mollenhauer et al., 2022; Kovachki et al., 2024; Adcock et al., 2024) studies the case where the input and output are elements in Banach spaces such as some function spaces. However, in our paper, although the objective is the distribution function, we are unable to directly observe the function itself. Instead, we can only observe one single data point sampled from this distribution. In other words, we do not have full feedback but only bandit feedback, which makes our problem much more difficult than general operator learning (Foster et al., 2018; Foster & Rakhlin, 2020).

**Functional Regression.** Functional data analysis focuses on analyzing data where each observation is a function defined over a continuous domain, typically sampled discretely from a population. Morris (2015); Ramsay & Silverman (2002) provide comprehensive introductions to functional regression. One subdomain, Kernel learning, for example, learning in reproducing kernel Hilbert space (RKHS) has been a hot topic in recent years (Yeh et al., 2023; Hou et al., 2023). Nonetheless, the problem we focus on is quite different from theirs, as datasets in kernel learning contain input and output points lying in some kernelized spaces, while we do not have access to data of required integral kernel $\mathcal{L}^{x,a}$ lying in an infinite-dimensional space $L^2(\Omega, \nu)$ directly. Zhang et al. (2022); Azizzadenesheli et al. studies functional regression for contextual CDFs in finite dimensions. For infinite dimensions, under some special conditions, Zhang et al. (2022) also proposes a method based on some regularization term which scales with respect to the number of data points.

## B. Useful Math Theorems

**Theorem B.1** (Theorem 8.2 in Gohberg et al. (2012))**.** *Let positive operator $\mathcal{U}(\theta)(w) = \int_\Omega \mathcal{K}(w, r)\theta(r)d\nu(r)$ be self-adjoint. If the kernel $K(w, r)$ is continuous and satisfies the Lipschitz condition,*

$$|K(w, r_1) - K(w, r_2)| \leq C||s_2 - s_1||,$$

*then*

$$\sum_{j=1}^\infty \lambda_j(\mathcal{U}) < \infty$$

*i.e., $\mathcal{U}$ is a trace class operator.*

**Theorem B.2** (Theorem 8.1 in Gohberg et al. (2012))**.** *If a positive Hilbert-Schmidt integral operator $\mathcal{U}$ is associated with kernel $\mathcal{K}(\cdot, \cdot)$. Then, if $\mathcal{U}$ is traceable, it holds that $tr(\mathcal{U}) = \int \mathcal{K}(x, x)dx < \infty$.*

**Theorem B.3** (Theorem 2.7 in Ferreira & Menegatto (2013))**.** *Assume $\nu$ is a Borel measure. Let $\mathcal{K}$ be a kernel in $C(\Omega \times \Omega) \cap L^2(\Omega \times \Omega, \nu \times \nu)$ possessing an integrable diagonal. Finally, assume $\mathcal{L}$ possesses a $L^2(\mathcal{X}, \nu)$-convergent spectral representation in the form*

$$\mathcal{L}(f) = \sum_{i=1}^\infty \lambda_i(\mathcal{L}) \langle f, e_i \rangle e_i, \ f \in L^2(\Omega, \nu),$$

*where $\{e_i\}$ is an orthonormal subset of $L^2(\Omega, \nu)$ and the sequence $\{\lambda_i(\mathcal{L})\}$ is a subset of a circle sector from the origin of $\mathbb{C}$ with central angle less than $\pi$. Then, the following statements hold:*

*(i) There exists $\xi \in [0, 2\pi]$ and $l > 0$ such that*

$$\sum_{i=1}^\infty |\lambda_i(\mathcal{L})| \leq (1 + l^2)^{1/2} \int_\Omega Re(e^{i\xi}K(x, x))d\nu(x);$$

*(ii) If the eigenvalues of $\mathcal{L}$ are arranged such that $|\lambda_i(\mathcal{L})| \geq |\lambda_{i+1}(\mathcal{L})|, i = 1, 2, \cdots$, then*

$$|\lambda_i(\mathcal{L})| \leq \frac{(1+l^2)^{1/2}}{n} \int_{\Omega} Re(e^{i\xi} K(x,x)) d\nu(x);$$

*(iii) as $n \to \infty$,*

$$\lambda_n(\mathcal{L}) = o(n^{-1}).$$

*(iv) The operator $\mathcal{L}$ is trace-class.*

**Definition B.4** (Functional Determinants). For any traceable Hilbert-Schmidt integral operator $\mathcal{U}$, we define the following number as the functional determinants of $\mathcal{U}$,

$$\mathcal{E}_{\mathcal{U}} \triangleq \sum_{i=1}^{\infty} \log(1 + \lambda_i(\mathcal{U})).$$

Given dataset $\mathcal{D}$, we will also use the terminology $\mathcal{E}_{\mathcal{D}}$ to denote $\mathcal{E}_{\mathcal{U}_{\mathcal{D}}}$ when it's clear from context. This number depicts the speed of growth of the eigenvalues of our integral operator. When the dimension is finite, one can show that the order is $\mathcal{E}_{\mathcal{D}} \simeq \log(|\mathcal{D}|)$.

**Theorem B.5** (Doob's Maximal Martingale Inequality). *Suppose $\{X_k\}_{k \geq 0}$ is a sub-martingale with $X_k \geq 0$ almost surely. Then for all $a > 0$, we have,*

$$\mathbb{P}(\max_{1 \leq i \leq k} X_i \geq a) \leq \frac{\mathbb{E}[X_k]}{a}.$$

**Lemma B.6.** *For any $a, b, c \in \mathbb{R}$, $a > 0$, we have*

$$\int_{\mathbb{R}} \exp\left\{-\frac{ax^2 + bx + c}{2}\right\} dx = \sqrt{\frac{2\pi}{a}} \exp\left(\frac{b^2 - 4ac}{8a}\right).$$

*Proof of Lemma B.6.* The proof is quite direct. By calculus, we have

$$-\frac{ax^2 + bx + c}{2} = -\frac{a}{2}\left(x^2 + \frac{b}{a}x + \frac{c}{a}\right) = -\frac{a}{2}\left(x + \frac{b}{2a}\right)^2 + \frac{b^2 - 4ac}{8a}.$$

Using the fact that $\int_{\mathbb{R}} e^{-x^2} = \sqrt{\pi}$, by change of variable $u = \sqrt{\frac{a}{2}}\left(x + \frac{b}{2a}\right)$, just plug it in and we shall finish the proof. $\square$

**Theorem B.7** (Courant-Fischer minimax theorem in Teschl (2014)). *Let $A$ be a compact, self-adjoint operator on a Hilbert space $H$. Its eigenvalues are listed in decreasing order*

$$\lambda_1 \geq \lambda_2 \geq \cdots \geq \lambda_k \geq \cdots.$$

*Let $S_k \subset H$ denote a $k$-dimensional subspace. Then,*

$$\lambda_k = \max_{S_k} \min_{\alpha \in S_k, ||\alpha||=1} \langle x, Ax \rangle.$$

**Theorem B.8** (Sion's Minimax Theorem in Sion (1958)). *Let $\mathcal{X}$ and $\mathcal{Y}$ be convex sets in linear topological spaces, and assume $\mathcal{X}$ is compact. Let $f : \mathcal{X} \times \mathcal{Y} \to \mathbb{R}$ be such that i) $f(x, \cdot)$ is concave and upper semicontinuous over $\mathcal{Y}$ for all $x \in \mathcal{X}$ and ii)$f(\cdot, y)$ is convex and lower semicontinuous over $\mathcal{X}$ for all $y \in \mathcal{Y}$. Then,*

$$\inf_{x \in \mathcal{X}} \sup_{y \in \mathcal{Y}} f(x,y) = \sup_{y \in \mathcal{Y}} \inf_{x \in \mathcal{X}} f(x,y).$$

**Lemma B.9** (Example 27.1 Shalev-Shwartz & Ben-David (2014)). *Suppose that $A$ is a compact set in $\mathbb{R}^m$, and set $c = \max_{a \in A} ||a||$, then $\mathcal{N}(r, A, ||\cdot||) \leq \left(\frac{2c\sqrt{m}}{r}\right)^m$*

## C. Proofs in Section 2

*Proof of Theorem 2.8.* Recall that

$$\mathcal{L}^{x,a}(\theta)(w) = \int_\Omega \mathcal{K}^{x,a}(w,r)\theta(r)d\nu(r).$$

By definition, we have $0 \leq \mathcal{K}^{x,a}(w,r) = \mathcal{K}^{x,a}(r,w) \leq 1$, and

$$\int_\Omega \int_\Omega \mathcal{K}^{x,a}(w,r)^2 d\nu(w)d\nu(r) \leq 1.$$

Thus, $\mathcal{L}^{x,a}$ is a Hilbert-Schmidt integral operator and it is compact. By Cauchy-Schwarz inequality, we have

$$||\mathcal{L}^{x,a}(\theta)||^2_{L^2(\Omega,\nu)} \leq \int_\Omega \int_\Omega \theta(r)^2 d\nu(r) \int_\Omega \mathcal{K}^{x,a}(w,r)^2 d\nu(r)d\nu(w) \leq ||\theta||_{L^2(\Omega,\nu)}.$$

Also,

$$\langle \theta, \mathcal{L}^{x,a}(\theta) \rangle = \int_\Omega \int_\Omega \theta(w)\mathcal{K}^{x,a}(w,r)\theta(r)d\nu(r)d\nu(w) \geq 0.$$

Therefore, $\mathcal{L}^{x,a}$ is positive operator with $||\mathcal{L}^{x,a}|| \leq 1$. By direct calculation, we have

$$\begin{aligned}
\langle \theta_1, \mathcal{L}^{x,a}\theta_2 \rangle &= \int_\Omega \theta_1(w) \int_\Omega \mathcal{K}^{x,a}(w,r)\theta_2(r)d\nu(r)d\nu(w) \\
&= \int_\Omega \theta_2(r) \int_\Omega \mathcal{K}^{x,a}(w,r)\theta_1(w)d\nu(w)d\nu(r) \\
&= \langle \mathcal{L}^{x,a}(\theta_1), \theta_2 \rangle.
\end{aligned}$$

So we finish the proof. $\square$

## D. Proofs in Section 3

*Proof of Theorem 3.4.* We prove this theorem for general $\varepsilon > 0$. We first show that $\sum_{j=1}^n \int_S \mathbb{I}_{y_j}(s)\phi(x_j,a_j,w,s)dm(s) \in L^2(\Omega,\nu)$. We have

$$\begin{aligned}
||\sum_{j=1}^n \int_S \mathbb{I}_{y_j}(s)\phi(x_j,a_j,w,s)dm(s)||^2_{L^2(\Omega,\nu)} &\leq \sum_{j=1}^n ||\int_S \mathbb{I}_{y_j}(s)\phi(x_j,a_j,w,s)dm(s)||^2_{L^2(\Omega,\nu)} \\
&\leq n\nu(\Omega) < \infty.
\end{aligned}$$

Now we define

$$\theta_0(w) \triangleq \mathcal{U}^\dagger_{\mathcal{D},\varepsilon}\left(\int_S \sum_{j=1}^n \mathbb{I}_{y_j}(s)\phi(x_j,a_j,w,s)dm(s)\right).$$

Because there are only finite terms in the summation by the definition of $\mathcal{U}^\dagger_{\mathcal{D},\varepsilon}$, we have

$$||\theta_0||_{L^2(\Omega,\nu)} < \infty.$$

Therefore, by the Cauchy-Schwarz inequality, we have

$$\begin{aligned}
&|\int_\Omega \theta_0(w)\phi(x_j,a_j,w,s)d\nu(w)| \\
&\leq \int_\Omega |\theta_0(w)\phi(x_j,a_j,w,s)|d\nu(w) \\
&\leq ||\theta_0||_{L^2(\Omega,\nu)}||\phi(x_j,a_j,\cdot,s)||_{L^2(\Omega,\nu)} \\
&\leq ||\theta_0||_{L^2(\Omega,\nu)}\sqrt{\nu(\Omega)}.
\end{aligned}$$

Recall we have $m(S) = 1 < \infty$ and $|\mathbb{I}_{y_j}(s)| \leq 1$ for any $j \in [n]$ and $s \in S$, which implies that

$$L(\theta_0, \mathcal{D}) < \infty.$$

On the other hand, for any $\theta \in \mathrm{Span}(e_1 \cdots, e_{N_\varepsilon})$, we have

$$L(\theta_0 + \theta, \mathcal{D}) - L(\theta_0, \mathcal{D}) = \sum_{j=1}^{n} \int_S \left( \int_\Omega \theta(w)\phi(x_j, a_j, w, s) \right)^2 dm(s)$$

$$+ 2\sum_{j=1}^{n} \int_S \left( \int_\Omega \theta(w)\phi(x)j, a_j, w, s)d\nu(w) \right) \left( \int_\Omega \theta_0(w)\phi(x_j, a_j, w, s)d\nu(w) - \mathbb{I}_{y_j}(s) \right) dm(s)$$

We analyze this difference term by term. First, by algebra, it holds that

$$\sum_{j=1}^{n} \int_S \left( \int_\Omega \theta(w)\phi(x_j, a_j, w, s)d\nu(w) \right) \left( \int_\Omega \theta_0(w)\phi(x_j, a_j, w, s)d\nu(w) \right) dm(s)$$

$$= \int_\Omega \theta(w) \int_\Omega \theta_0(r) \left( \sum_{j=1}^{n} \int_S \phi(x_j, a_j, w, s)\phi(x_j, a_j, r, s)dm(s) \right) d\nu(w)d\nu(r)$$

$$= \int_\Omega \theta(w) \sum_{j=1}^{n} \mathcal{L}^{x_j, a_j}(\theta_0)(w)d\nu(w)$$

$$= \langle \theta, \mathcal{U}_\mathcal{D}(\theta_0) \rangle.$$

Recall $\theta, \theta_0 \in \mathrm{Span}(e_1 \cdots, e_{N_\varepsilon})$, so we have

$$\langle \theta, \mathcal{U}_\mathcal{D}(\theta_0) \rangle = \left\langle \theta, \widehat{\mathcal{U}}_{\mathcal{D}, \varepsilon}(\theta_0) \right\rangle.$$

Second, by direct calculation, we also have

$$\sum_{j=1}^{n} \int_S \left( \int_\Omega \theta(w)\phi(x_j, a_j, w, s)d\nu(w) \right) \mathbb{I}_{y_j}(s)dm(s)$$

$$= \int_\Omega \theta(w) \left( \sum_{j=1}^{n} \int_S \mathbb{I}_{y_j}(s)\phi(x_j, a_j, w, s)dm(s) \right) d\nu(w)$$

$$= \left\langle \theta, \sum_{j=1}^{n} \int_S \mathbb{I}_{y_j}(s)\phi(x_j, a_j, \cdot, s)dm(s) \right\rangle$$

Given the condition that $\theta \in \mathrm{Span}(e_1 \cdots, e_{N_\varepsilon})$, we have

$$\left\langle \theta, \sum_{j=1}^{n} \int_S \mathbb{I}_{y_j}(s)\phi(x_j, a_j, \cdot, s)dm(s) \right\rangle = \sum_{i=1}^{N_\varepsilon} \langle \theta, e_i \rangle \left\langle \sum_{j=1}^{n} \int_S \mathbb{I}_{y_j}(s)\phi(x_j, a_j, \cdot, s)dm(s), e_i \right\rangle$$

Moreover, by the definition of $\theta_0$, we know that

$$\left\langle \theta, \widehat{\mathcal{U}}_{\mathcal{D}, \varepsilon}(\theta_0) \right\rangle = \sum_{i=1}^{N_\varepsilon} \langle \theta, e_i \rangle \left\langle \lambda_i \cdot \frac{1}{\lambda_i} \sum_{j=1}^{n} \int_S \mathbb{I}_{y_j}(s)\phi(x_j, a_j, \cdot, s)dm(s), e_i \right\rangle$$

$$= \sum_{i=1}^{N_\varepsilon} \langle \theta, e_i \rangle \left\langle \sum_{j=1}^{n} \int_S \mathbb{I}_{y_j}(s)\phi(x_j, a_j, \cdot, s)dm(s), e_i \right\rangle.$$

Thus,

$$L(\theta_0 + \theta, \mathcal{D}) - L(\theta_0, \mathcal{D}) = \sum_{j=1}^{n} \int_S \left( \int_\Omega \theta(w) \phi(x_j, a_j, w, s) \right)^2 dm(s) \geq 0.$$

Therefore, $\theta_0 = \theta_{\mathcal{D},\varepsilon}$ and we finish the proof. $\qquad\square$

**Lemma D.1.** *Projection $\mathcal{P}_\mathcal{D}$ is non-expansive, which implies that*

$$||\mathcal{P}_\mathcal{D}(x) - \mathcal{P}_\mathcal{D}(z)||_{\mathcal{U}_\mathcal{D}} \leq ||x - z||_{\mathcal{U}_\mathcal{D}}, \ \forall x, z \in \mathcal{L}^2(\Omega, \nu).$$

*Proof of Lemma D.1.* By the fact that $\mathcal{C}$ is convex, we define the quadratic function $h(\mu) = ||x - ((1-\mu)\mathcal{P}_\mathcal{D}(x) + \mu y)||^2_{\mathcal{U}_\mathcal{D}}$ for any $x \in L^2(\Omega, \nu), y \in \mathcal{C}$. We expand it to get,

$$h(\mu) = ||(x - \mathcal{P}_\mathcal{D}(x)) - \mu(y - \mathcal{P}_\mathcal{D}(x))||^2_{\mathcal{U}_\mathcal{D}} = ||x - \mathcal{P}_\mathcal{D}(x)||^2_{\mathcal{U}_\mathcal{D}} - 2\mu \langle x - \mathcal{P}_\mathcal{D}(x), \mathcal{U}_\mathcal{D}(y - \mathcal{P}_\mathcal{D}(x)) \rangle + \mu^2 ||y - \mathcal{P}_\mathcal{D}(x)||^2_{\mathcal{U}_\mathcal{D}}.$$

The vertex of $h(\mu)$ is

$$\mu^* = \frac{\langle x - \mathcal{P}_\mathcal{D}(x), \mathcal{U}_\mathcal{D}(y - \mathcal{P}_\mathcal{D}(x)) \rangle}{||y - \mathcal{P}_\mathcal{D}(x)||^2_{\mathcal{U}_\mathcal{D}}}.$$

Because $\mathcal{P}_\mathcal{D}(x) = \text{argmin}_{y \in \mathcal{C}} ||y - x||^2_{\mathcal{U}_\mathcal{D}}$, we have $\mu^* \leq 0$. Thus,

$$\langle x - \mathcal{P}_\mathcal{D}(x), \mathcal{U}_\mathcal{D}(y - \mathcal{P}_\mathcal{D}(x)) \rangle \leq 0.$$

Again, we abbreviate $\lambda_i$ as $\lambda_i(\mathcal{U}_\mathcal{D})$ and $e_i$ as $e^i_{\mathcal{U}_\mathcal{D}}$. By the spectral representation of $\mathcal{U}_\mathcal{D}$, this is equivalent to

$$\sum_{i=1}^{\infty} \lambda_i \langle x - \mathcal{P}_\mathcal{D}(x), e_i \rangle \langle y - \mathcal{P}_\mathcal{D}(x), e_i \rangle \leq 0. \tag{2}$$

This holds for all $y \in \mathcal{C}$, so we set $y = \mathcal{P}_\mathcal{D}(z)$ in Equation (2) to get

$$\sum_{i=1}^{\infty} \lambda_i \langle x - \mathcal{P}_\mathcal{D}(x), e_i \rangle \langle \mathcal{P}_\mathcal{D}(z) - \mathcal{P}_\mathcal{D}(x), e_i \rangle \leq 0. \tag{3}$$

On the other hand, we switch $x, z$ in Equation (3) to get,

$$\sum_{i=1}^{\infty} \lambda_i \langle \mathcal{P}_\mathcal{D}(z) - z, e_i \rangle \langle \mathcal{P}_\mathcal{D}(z) - \mathcal{P}_\mathcal{D}(x), e_i \rangle \leq 0. \tag{4}$$

Adding Equation (4) and Equation (3) to obtain

$$\sum_{i=1}^{\infty} \lambda_i \langle \mathcal{P}_\mathcal{D}(z) - \mathcal{P}_\mathcal{D}(x), e_i \rangle \langle \mathcal{P}_\mathcal{D}(z) - \mathcal{P}_\mathcal{D}(x), e_i \rangle \leq \sum_{i=1}^{\infty} \lambda_i \langle z - x, e_i \rangle \langle \mathcal{P}_\mathcal{D}(z) - \mathcal{P}_\mathcal{D}(x), e_i \rangle.$$

By Cauchy-Schwartz inequality, we have,

$$\sum_{i=1}^{\infty} \lambda_i \langle z - x, e_i \rangle \langle \mathcal{P}_\mathcal{D}(z) - \mathcal{P}_\mathcal{D}(x), e_i \rangle \leq \left( \sum_{i=1}^{\infty} \lambda_i \langle z - x, e_i \rangle^2 \right)^{1/2} \left( \sum_{i=1}^{\infty} \lambda_i \langle \mathcal{P}_\mathcal{D}(z) - \mathcal{P}_\mathcal{D}(x), e_i \rangle^2 \right)^{1/2}.$$

Plugging this back, we get

$$(\sum_{i=1}^{\infty} \lambda_i \langle \mathcal{P}_\mathcal{D}(z) - \mathcal{P}_\mathcal{D}(x), e_i \rangle^2)^{1/2} \leq \left( \sum_{i=1}^{\infty} \lambda_i \langle z - x, e_i \rangle^2 \right)^{1/2},$$

which is exactly $||\mathcal{P}_\mathcal{D}(x) - \mathcal{P}_\mathcal{D}(z)||_{\mathcal{U}_\mathcal{D}} \leq ||x - z||_{\mathcal{U}_\mathcal{D}}$. So we finish the proof. $\qquad\square$

*Proof of Theorem 3.6.* In this section, we also first prove for a general $\varepsilon$, and then we replace $\varepsilon$ by $\varepsilon^* = n^{-\frac{2}{\gamma+2}}$ to obtain our result. For the ease of notation, we use $\lambda_i$ to denote $\lambda_i(\mathcal{U}_{\mathcal{D}})$ and $e_i$ to denote $e_{\mathcal{U}_{\mathcal{D}}}^i$. We first compute the difference

$$
\begin{aligned}
&\theta_{\mathcal{D},\varepsilon} - \theta^* \\
=&\mathcal{U}_{\mathcal{D},\varepsilon}^{\dagger}\left(\int_S \sum_{j=1}^n \mathbb{I}_{y_j}(s)\phi(x_j,a_j,w,s)dm(s)\right) - \sum_{i=1}^{\infty}\langle\theta^*,e_i\rangle e_i \\
=&\mathcal{U}_{\mathcal{D},\varepsilon}^{\dagger}\left(\int_S \sum_{j=1}^n \mathbb{I}_{y_j}(s)\phi(x_j,a_j,w,s)dm(s)\right) - \mathcal{U}_{\mathcal{D},\varepsilon}^{\dagger}(\mathcal{U}_{\mathcal{D}}(\theta^*)) - \sum_{i\geq N_\varepsilon+1}\langle\theta^*,e_i\rangle e_i \\
=&\mathcal{U}_{\mathcal{D},\varepsilon}^{\dagger}\left(\int_S \sum_{j=1}^n \mathbb{I}_{y_j}(s)\phi(x_j,a_j,w,s)dm(s) - \int_S\int_\Omega \theta^*(r)\phi(x_j,a_j,r,s)d\nu(r)\phi(x_j,a_j,w,s)dm(s)\right) - \sum_{i\geq N_\varepsilon+1}\langle\theta^*,e_i\rangle e_i.
\end{aligned}
\tag{5}
$$

We define

$$
V_j(w) \triangleq \int_S \mathbb{I}_{y_j}(s)\phi(x_j,a_j,w,s)dm(s) - \int_S\int_\Omega \theta^*(r)\phi(x_j,a_j,r,s)d\nu(r)\phi(x_j,a_j,w,s)dm(s),
$$

$$
W_n \triangleq \sum_{j=1}^n V_j.
$$

We apply norm $||\cdot||_{\mathcal{U}_{\mathcal{D}}}$ on both sides of Equation (5) yielding

$$
\begin{aligned}
||\theta_{\mathcal{D},\varepsilon} - \theta^*||_{\mathcal{U}_{\mathcal{D}}} &\leq ||\mathcal{U}_{\mathcal{D},\varepsilon}^{\dagger}(W_n)||_{\mathcal{U}_{\mathcal{D}}} + ||\sum_{i\geq N_\varepsilon+1}\langle\theta^*,e_i\rangle e_i||_{\mathcal{U}_{\mathcal{D}}} \\
&= ||W_n||_{\widehat{\mathcal{U}}_{\mathcal{D},\varepsilon}^{\dagger}} + ||\sum_{i\geq N_\varepsilon+1}\langle\theta^*,e_i\rangle e_i||_{\mathcal{U}_{\mathcal{D}}} \\
&\leq ||W_n||_{\widehat{\mathcal{U}}_{\mathcal{D},\varepsilon}^{\dagger}} + n\varepsilon M
\end{aligned}
\tag{6}
$$

The equality is due to the definition of $\widehat{\mathcal{U}}_{\mathcal{D},\varepsilon}^{\dagger}$. Hence, our analysis only needs to focus on the first term $||W_n||_{\widehat{\mathcal{U}}_{\mathcal{D},\varepsilon}^{\dagger}}$. We use $\mathcal{F}_k$ to denote the $\sigma$-algebra generated by all the data from $(x_1,a_1,y_1)$ to $(x_k,a_k,y_k)$, and $\mathcal{F}_\infty$ is $\sigma(\cup_k\mathcal{F}_k)$ correspondingly. By the definition of $V_j$, we take expectation with respect to $\mathcal{F}_{j-1}$ and obtain

$$
\mathbb{E}[V_j(w)|\mathcal{F}_{j-1}] = \int_S\int_\Omega \theta^*(r)\phi(x_j,a_j,r,s)d\nu(r)\phi(x_j,a_j,w,s)dm(s) - \theta^*(r)\phi(x_j,a_j,r,s)d\nu(r)\phi(x_j,a_j,w,s)dm(s) = 0.
$$

Therefore, $\{V_j\}_{j\geq 1}$ is a martingale difference sequence and consequently, $W_k = \sum_{j=1}^k V_j$ is a martingale. For any $\alpha \in L^2(\Omega,\nu)$, we define $M_k(\alpha) \triangleq \exp\{\langle\alpha,W_k\rangle - \frac{1}{2}||\alpha||_{\mathcal{U}_{\mathcal{D}}}^2\}$. Moreover, we define $M_0(\alpha) = 1$ for any $\alpha$. We have the following Lemma D.2 that $\{M_k(\alpha)\}_{k\geq 0}$ is a non-negative supermartingale.

**Lemma D.2.** $\{M_k(\alpha)\}_{k\geq 0}$ *is a non-negative supermartingale.*

Define $\zeta = \{\zeta_i\}_{i=1}^{\infty}$ as an infinite sequence of i.i.d. standard Gaussian random variables. We use $\mathcal{F}^\zeta$ to denote the $\sigma(\zeta_1,\zeta_2,\cdots)$. Define

$$
\beta \triangleq \sum_{i=1}^{N_\varepsilon}\zeta_i e_i,
$$

Because there are only finite terms in the summation, it holds that

$$
\mathbb{E}[||\beta||_{L^2(\Omega,\nu)}^2] < \infty.
$$

Thus, we have $||\beta||_{L^2(\Omega,\nu)} < \infty$ a.s. and $\beta \in L^2(\Omega, \nu)$.

Denoting $\overline{M}_k$ as $\mathbb{E}[M_k(\beta)|\mathcal{F}_\infty]$, we now verify that this is also a non-negative supermartingale.

$$
\begin{aligned}
\mathbb{E}[\overline{M}_k|\mathcal{F}_{k-1}] &= \mathbb{E}[M_k(\beta)|\mathcal{F}_{k-1}] \\
&= \mathbb{E}[\mathbb{E}[M_k(\beta)|\mathcal{F}^\zeta, \mathcal{F}_{k-1}]|\mathcal{F}_{k-1}] \\
&\leq \mathbb{E}[\mathbb{E}[M_{k-1}(\beta)|\mathcal{F}^\zeta, \mathcal{F}_{k-1}]|\mathcal{F}_{k-1}] \\
&= \mathbb{E}[\mathbb{E}[M_{k-1}(\beta)|\mathcal{F}^\zeta, \mathcal{F}_\infty]|\mathcal{F}_{k-1}] \\
&= \mathbb{E}[\mathbb{E}[M_{k-1}(\beta)|\mathcal{F}_\infty]|\mathcal{F}_{k-1}] \\
&= \overline{M}_{k-1}.
\end{aligned}
$$

Furthermore, the integral of $|\overline{M}_n|$ could be upper bounded by,

$$
\mathbb{E}[|\overline{M}_k|] = \mathbb{E}[\overline{M}_n] = \mathbb{E}[M_k(\beta)] = \mathbb{E}[\mathbb{E}[M_k(\beta)|\mathcal{F}^\zeta]] \leq \mathbb{E}[M_0(\beta)] = 1.
$$

Therefore, $\{\overline{M}_k\}_{n \geq k \geq 0}$ is a non-negative supermartingale with a uniform expectation upper bound.

On the other hand, we can directly calculate $\overline{M}_n$. We use $w_i$ to denote $\langle W_n, e_i \rangle$ for simplicity. Then, it holds that

$$
M_n(\beta) = \exp\left\{ \langle \beta, W_n \rangle - \frac{1}{2}||\beta||^2_{\mathcal{U}_\mathcal{D}} \right\} = \exp\left\{ \sum_{i=1}^{N_\varepsilon} w_i \zeta_i - \frac{1}{2}\sum_{i=1}^{N_\varepsilon} \lambda_i \zeta_i^2 \right\},
$$

and

$$
\begin{aligned}
\overline{M}_n &= \int_{\mathbb{R}^{N_\varepsilon}} \exp\left\{ \sum_{i=1}^{N_\varepsilon} w_i \zeta_i - \frac{1}{2}\sum_{i=1}^{N_\varepsilon} \left(\lambda_i \zeta_i^2 + \zeta_i^2\right) \right\} \frac{1}{\sqrt{2\pi}}d\zeta_1 \frac{1}{\sqrt{2\pi}}d\zeta_2 \cdots \frac{1}{\sqrt{2\pi}}d\zeta_{N_\varepsilon} \\
&= \prod_{i=1}^{N_\varepsilon} \left( \int_{\mathbb{R}} \exp\left\{ w_i \zeta_i - \frac{\lambda_i + 1}{2}\zeta_i^2 \right\} \frac{1}{\sqrt{2\pi}}d\zeta_i \right).
\end{aligned}
$$

Then, by Lemma B.6, we have

$$
\overline{M}_n = \frac{1}{\sqrt{\prod_{i=1}^{N_\varepsilon}(1+\lambda_i)}} \exp\left\{ \sum_{i=1}^{N_\varepsilon} \frac{w_i^2}{2(1+\lambda_i)} \right\} \geq \frac{1}{\sqrt{\prod_{i=1}^{N_\varepsilon}(1+\lambda_i)}} \exp\left\{ \frac{1}{2(1+1/(n\varepsilon))}||W_n||^2_{\mathcal{U}^\dagger_{\mathcal{D},\varepsilon}} \right\}.
$$

Applying Theorem B.5, we derive

$$
\mathbb{P}\left( \frac{1}{\sqrt{\prod_{i=1}^{N_\varepsilon}(1+\lambda_i)}} \exp\left\{ \frac{1}{2(1+1/(n\varepsilon))}||W_n||^2_{\mathcal{U}^\dagger_{\mathcal{D},\varepsilon}} \right\} \geq \frac{1}{\delta} \right) \leq \delta,
$$

which implies that with probability $1 - \delta$,

$$
||W_n||^2_{\mathcal{U}^\dagger_{\mathcal{D},\varepsilon}} < 2(1+1/(n\varepsilon))\left[ \log(\frac{1}{\delta}) + \frac{1}{2}\sum_{i=1}^{N_\varepsilon} \log(1+\lambda_i) \right].
$$

Combined with Equation (6), it indicates that with probability $1 - \delta$,

$$
||\theta_{\mathcal{D},\varepsilon} - \theta^*||_{\mathcal{U}_\mathcal{D}} \leq \sqrt{2(1+1/(n\varepsilon))\left[ \log(\frac{1}{\delta}) + \frac{1}{2}\sum_{i=1}^{N_\varepsilon} \log(1+\lambda_i) \right]} + n\varepsilon M.
$$

Noting that projection mapping is non-expansive by Lemma D.1, we have

$$
||\mathcal{P}_\mathcal{D}(\theta_{\mathcal{D},\varepsilon}) - \mathcal{P}_\mathcal{D}(\theta^*)||_{\mathcal{U}_\mathcal{D}} = ||\widehat{\theta}_{\mathcal{D},\varepsilon} - \theta^*||_{\mathcal{U}_\mathcal{D}} \leq ||\theta_{\mathcal{D},\varepsilon} - \theta^*||_{\mathcal{U}_\mathcal{D}}.
$$

Finally, we conclude that

$$||\widehat{\theta}_{\mathcal{D},\varepsilon} - \theta^*||_{\mathcal{U}_{\mathcal{D}}} \leq \sqrt{(1 + 1/n\varepsilon) \left(2\log(\frac{1}{\delta}) + \sum_{i=1}^{N_\varepsilon} \log(1 + \lambda_i)\right)} + n\varepsilon M.$$

We define

$$\mathcal{E}_{\mathcal{D},\varepsilon}^{\delta}(n) \triangleq \sqrt{2\left(2\log(\frac{1}{\delta}) + \sum_{i=1}^{N_\varepsilon} \log(1 + \lambda_i)\right)} + n\varepsilon M.$$

Setting $\varepsilon = \varepsilon^* = n^{-\frac{2}{\gamma+2}}$ and notice that $\frac{1}{n\varepsilon} = n^{-\frac{\gamma}{\gamma+2}} \leq 1$, So we finish the proof. $\qquad\square$

*Proof of Lemma D.2.* By definition, we know that $M_k(\alpha)$ is $\mathcal{F}_k$-measurable. Consequently, it holds that

$$\mathbb{E}\left[M_k(\alpha)|\mathcal{F}_{k-1}\right] M_{k-1}(\alpha) = \mathbb{E}\left[\exp\left\{\alpha, V_k\right\} - \frac{1}{2}\int_S \psi_{x_k,a_k}(\alpha,t)^2 dm(s)|\mathcal{F}_{k-1}\right]$$

$$= M_{k-1}(\alpha) \frac{\mathbb{E}\left[\exp\left\{\alpha, V_k\right\}|\mathcal{F}_{k-1}\right]}{\exp\left\{\frac{1}{2}\int_S \psi_{x_k,a_k}(\alpha,t)^2 dm(s)\right\}}$$

Since $-\int_S |\psi_{x_k,a_k}(\alpha,t)| dm(s) \leq \langle \alpha, V_k \rangle \leq \int_S |\psi_{x_k,a_k}(\alpha,t)| dm(s)$, according to Hoeffding's lemma (Hoeffding, 1994), we have

$$\mathbb{E}\left[\exp\left\{\alpha, V_k\right\}|\mathcal{F}_{k-1}\right] \leq \exp\left\{\frac{4}{8}\left(\int_S |\psi_{x_k,a_k}(\alpha,t)| dm(s)\right)^2\right\} \leq \exp\left\{\frac{1}{2}\int_S \psi_{x_k,a_k}(\alpha,t)^2 dm(s)\right\}.$$

Then we have

$$\mathbb{E}\left[M_k(\alpha)|\mathcal{F}_{k-1}\right] \leq M_{k-1}(\alpha) \frac{\exp\left\{\frac{1}{2}\int_S \psi_{x_k,a_k}(\alpha,t)^2 dm(s)\right\}}{\exp\left\{\frac{1}{2}\int_S \psi_{x_k,a_k}(\alpha,t)^2 dm(s)\right\}} = M_{k-1}(\alpha).$$

We finish the proof. $\qquad\square$

*Proof of Lemma 3.7.* For any $k$ fixed, by Theorem B.7, we have,

$$\lambda_k(\mathcal{U}_{\mathcal{D}}) = \max_{S_k,\dim(S_k)=k} \min_{\alpha\in S_k,||\alpha||=1} \langle \alpha, (\mathcal{L}^{x_1,a_1} + \cdots + \mathcal{L}^{x_n,a_n})(\alpha)\rangle$$

$$= \max_{S_k,\dim(S_k)=k} \min_{\alpha\in S_k,||\alpha||=1} \sum_{j=1}^{n} \langle \alpha, \mathcal{L}^{x_j,a_j}(\alpha)\rangle$$

$$\leq \max_{S_k,\dim(S_k)=k} \min_{\alpha\in S_k,||\alpha||=1} \max_{\mathcal{L}\in\{\mathcal{L}^{x,a}:x\in\mathcal{X},a\in\mathcal{A}\}} n\langle \alpha, \mathcal{L}(\alpha)\rangle$$

Notice that for any $S_k$ fixed, the set $\{\alpha\in S_k, ||\alpha||=1\}$ is both compact and convex. By Assumption 2.10, we also know that $\{\mathcal{L}^{x,a}: x\in\mathcal{X}, a\in\mathcal{A}\}$ is convex. We regard $\langle\alpha,\mathcal{L}(\alpha)\rangle$ as a function of both $\alpha$ and $\mathcal{L}$. Then, function $f(\alpha,\mathcal{L}) \triangleq \langle\alpha,\mathcal{L}(\alpha)\rangle$ is linear in $\mathcal{L}$, and so $f(\alpha,\mathcal{L})$ is continuous and concave in $\mathcal{L}$. On the other hand, $f(\alpha,\mathcal{L})$ is the square of some norm $||\cdot||_{\mathcal{L}}$ and thus it is convex and continuous in $\alpha$. We apply Theorem B.8 to get

$$\max_{S_k,\dim(S_k)=k} \min_{\alpha\in S_k,||\alpha||=1} \max_{\mathcal{L}\in\{\mathcal{L}^{x,a}:x\in\mathcal{X},a\in\mathcal{A}\}} \langle\alpha,\mathcal{L}(\alpha)\rangle$$

$$= \max_{S_k,\dim(S_k)=k} \max_{\mathcal{L}\in\{\mathcal{L}^{x,a}:x\in\mathcal{X},a\in\mathcal{A}\}} \min_{\alpha\in S_k,||\alpha||=1} \langle\alpha,\mathcal{L}(\alpha)\rangle$$

$$= \max_{\mathcal{L}\in\{\mathcal{L}^{x,a}:x\in\mathcal{X},a\in\mathcal{A}\}} \max_{S_k,\dim(S_k)=k} \min_{\alpha\in S_k,||\alpha||=1} \langle\alpha,\mathcal{L}(\alpha)\rangle.$$

The last inequality holds because we can swap the order of two supremums. Using Theorem B.7 once again, we have

$$\max_{\mathcal{L}\in\{\mathcal{L}^{x,a}:x\in\mathcal{X},a\in\mathcal{A}\}} \max_{S_k,\dim(S_k)=k} \min_{\alpha\in S_k,||\alpha||=1} \langle\alpha,\mathcal{L}(\alpha)\rangle = \max_{\mathcal{L}\in\{\mathcal{L}^{x,a}:x\in\mathcal{X},a\in\mathcal{A}\}} \lambda_k(\mathcal{L}).$$

Combining these parts together, it holds that

$$\lambda_k(\mathcal{U}_{\mathcal{D}}) \leq n \max_{\mathcal{L} \in \{\mathcal{L}^{x,a}: x \in \mathcal{X}, a \in \mathcal{A}\}} \lambda_k(\mathcal{L}) \leq n\tau_i.$$

Thus, we have

$$\sum_{i=1}^{N_\varepsilon} \log(1 + \lambda_i(\mathcal{U}_{\mathcal{D}})) = \sum_{i=1}^{N_\varepsilon} \log(1 + \frac{\lambda_i}{n\varepsilon} n\varepsilon) \leq \sum_{i=1}^{N_\varepsilon} \log(1 + \frac{n\tau_i}{n\varepsilon} n\varepsilon)$$

Because for $\forall 1 \leq i \leq N_\varepsilon$, it holds that

$$n\varepsilon \leq \lambda_i \leq n\tau_i,$$

we have

$$\sum_{i=1}^{N\varepsilon} \log(1 + \frac{n\tau_i}{n\varepsilon} n\varepsilon) = \sum_{i=1}^{N_\varepsilon} \log(1 + n\tau_i) \leq \sum_{i=1}^{N_\varepsilon} \log(1 + n\tau_i^{1-\gamma} \varepsilon^\gamma \frac{\tau_i^\gamma}{\varepsilon^\gamma}) \leq \sum_{i=1}^{N_\varepsilon} \frac{\tau_i^\gamma}{\varepsilon^\gamma} \log(1 + n\tau_i^{1-\gamma} \varepsilon^\gamma) \leq \frac{\log(1 + \varepsilon^\gamma n)}{\varepsilon^\gamma} \sum_{i=1}^{N_\varepsilon} \tau_i,$$

which implies

$$\sum_{i=1}^{N_\varepsilon} \log(1 + \lambda_i(\mathcal{U}_{\mathcal{D}})) \leq \frac{\log(1 + \varepsilon^\gamma n)}{\varepsilon^\gamma} \sum_{i=1}^{N_\varepsilon} \tau_i. \tag{7}$$

Plugging Equation (7) back to the definition of $\mathcal{E}_{\mathcal{D},\varepsilon}^\delta(n)$, we have,

$$\mathcal{E}_{\mathcal{D},\varepsilon}^\delta(n) = \sqrt{2\left(2\log(\frac{1}{\delta}) + \sum_{i=1}^{N_\varepsilon} \log(1 + \lambda_i)\right) + n\varepsilon M}$$

$$\leq \sqrt{2\left(2\log(\frac{1}{\delta}) + \frac{\log(1 + \varepsilon^\gamma n)}{\varepsilon^\gamma} \sum_{i=1}^{N_\varepsilon} \tau_i\right) + n\varepsilon M}$$

$$\leq \sqrt{2\left(2\log(\frac{1}{\delta}) + \frac{\log(1 + \varepsilon^\gamma n)}{\varepsilon^\gamma} s_0\right) + n\varepsilon M}.$$

Setting $\varepsilon^* = n^{-\frac{2}{\gamma+2}}$, we get

$$\mathcal{E}_{\mathcal{D},\varepsilon^*}^\delta(n) \leq \sqrt{4\log(1/\delta) + 2s_0 n^{\frac{2\gamma}{\gamma+2}} \log(1+n)} + n^{\frac{\gamma}{\gamma+2}} M.$$

The RHS has nothing to do with $\mathcal{D}$ but the information $|\mathcal{D}| = n$. Therefore, we define $\mathcal{E}_\delta(n) \triangleq 2\log(1/\delta)^{1/2} + \left(2\sqrt{s_0 \log(1+n)} + M\right) n^{\frac{\gamma}{\gamma+2}}$. By the fact that $\sqrt{a+b} \leq \sqrt{a} + \sqrt{b}$, we obtain,

$$\mathcal{E}_{\mathcal{D},\varepsilon^*}^\delta(n) \leq \mathcal{E}_\delta(n) = 2\log(1/\delta)^{1/2} + \left(2\sqrt{s_0 \log(1+n)} + M\right) n^{\frac{\gamma}{\gamma+2}}.$$

Thus, we finish the proof. $\qquad\square$

*Proof of Theorem 3.10.* Denoting $\mathcal{K}^{x,a}(w,r)$ as $g_{w,r}(x,a)$, we get a function family indexed by $\Omega \times \Omega$, $\mathcal{G} = \{g_{w,r}(x,a) : (w,r) \in \Omega \times \Omega\}$. We first consider all the rational points in $\Omega \times \Omega$ which forms a countable subset $\{(w_q, r_q)\}_{q=1}^\infty$ and induces a countable subset of $\mathcal{G}_\mathbb{Q} = \{g_{w_q,r_q}(x,a)\}_{q=1}^\infty$. By Theorem 3.4.5 in Giné & Nickl (2021), we have, for any $\delta > 0$,

$$\mathbb{P}\left(\sup_q |g_{w_q,r_q}(x,a) - \int_{\mathcal{X},\mathcal{A}} g_{w_q,r_q}(x,a)dQ| \geq 2\mathbb{E}\left[\sup_q \frac{1}{n} \sum_{j=1}^n \epsilon_j g_{w_q,r_q}(x_j,a_j)\right] + \sqrt{\frac{2\log(1/\delta)}{n}}\right) \leq \delta.$$

By the density of rational numbers in real numbers and $g$ is $2L_0$-Lipschitz continuous in $(w, r)$, we know that

$$\mathbb{P}\left(\sup_{g \in \mathcal{G}} \left| g_{w,r}(x, a) - \int_{\mathcal{X}, \mathcal{A}} g_{w,r}(x, a) dQ \right| \geq 2\mathbb{E}\left[\sup_{g \in \mathcal{G}} \frac{1}{n} \sum_{j=1}^{n} \epsilon_j g_{w,r}(x_j, a_j)\right] + \sqrt{\frac{2 \log(1/\delta)}{n}}\right) \leq \delta.$$

Now we try to bound the Rademacher complexity term $\mathbb{E}\left[\sup_{g \in \mathcal{G}} \frac{1}{n} \sum_{j=1}^{n} \epsilon_j g_{w,r}(x_j, a_j)\right]$. Recall that we have $\mathcal{N}(t, \Omega^2, \|\cdot\|_\infty) \leq (\frac{A}{t})^{2d}$, $\|g_{w_1, r_1}(x, a) - g_{w_2, r_2}(x, a)\|_\infty \leq 2L_0 \|(w_1, r_1) - (w_2, r_2)\|_\infty$. Then,

$$\mathcal{N}(t, \mathcal{G}, \|\cdot\|_\infty) \leq \left(\frac{2L_0 A}{t}\right)^{2d}.$$

Then, by Dudley's integral entropy bound and using $\|g\|_\infty \leq 1$, we have

$$\mathbb{E}_\varepsilon \sup_{g \in \mathcal{G}} \frac{1}{n} \sum_{j=1}^{n} \epsilon_j g(x_j, a_j) \leq \frac{12}{\sqrt{n}} \int_0^1 \sqrt{\log \mathcal{N}(y, \|\cdot\|_\infty, \mathcal{G})} dy \leq \frac{24\sqrt{d}}{\sqrt{n}} \sqrt{\log(2L_0 A)}.$$

For the simplicity of notation, we use $Q(\mathcal{K}^{x,a}(w, r))$ to denote $\int_{\mathcal{X}, \mathcal{A}} \int_S \phi(x, a, w, s) \phi(x, a, r, s) dm(s) dQ$ and $Q_n(\mathcal{K}^{x,a}(w, r))$ to denote its empirical counterpart. Combining all the parts above together, we have that with probability at least $1 - \delta$,

$$\sup_{(w,r)} |Q(\mathcal{K}^{x,a}(w, r)) - Q_n(\mathcal{K}^{x,a}(w, r))| \leq \frac{48\sqrt{d \log(2L_0 A)}}{\sqrt{n}} + \frac{\sqrt{2 \log(1/\delta)}}{\sqrt{n}}.$$

By Assumption 3.9, we get with probability at least $1 - \delta$,

$$\sup_{(w,r)} \left| \frac{Q(\mathcal{K}^{x,a}(w, r))}{Q_n(\mathcal{K}^{x,a}(w, r))} - 1 \right| \leq \frac{48\sqrt{d \log(2L_0 A)}}{\eta\sqrt{n}} + \frac{\sqrt{2 \log(1/\delta)}}{\eta\sqrt{n}}.$$

Recalling that $\mathcal{U}_\mathcal{D} = \sum_{j=1}^{n} \int_\Omega \mathcal{K}^{x_j, a_j}(w, r) dm(s) d\nu(r)$, our bound in Corollary 3.8 yields with probability at least $1 - \delta$, the following bound holds that

$$\|\widehat{\theta}_\mathcal{D} - \theta^*\|_{\mathcal{U}_\mathcal{D}} \leq \mathcal{E}_\delta(n).$$

We use $\mathcal{U}^*$ to denote the integral operator induced by kernel $n \int_{\mathcal{X}, \mathcal{A}} \int_S \phi(x, a, w, t) \phi(x, a, r, t) dm(s) dQ_{x,a}$ and $\mathcal{U}_\mathcal{D}$ is its empirical counterpart. Therefore, for any $\theta$, we have that

$$\langle \theta, \mathcal{U}^*(\theta) \rangle = \int_\Omega \theta(w) \int_\Omega n \int_{\mathcal{X}, \mathcal{A}} \int_S \theta(r) \phi(x, a, w, t) \phi(x, a, r, t) dm(s) dQ_{x,a} d\nu(r) d\nu(w)$$

$$= n \int_\Omega \int_\Omega \theta(w) \theta(r) Q(\mathcal{K}(w, r)) d\nu(w) d\nu(r),$$

$$\langle \theta, \mathcal{U}_\mathcal{D}(\theta) \rangle = \int_\Omega \theta(w) \int_\Omega \sum_{j=1}^{n} \int_S \theta(r) \phi(x_j, a_j, w, s) \phi(x_j, a_j, r, s) dm(s) d\nu(w) d\nu(r)$$

$$= n \int_\Omega \int_\Omega \theta(w) \theta(r) Q_n(\mathcal{K}(w, r)) d\nu(w) d\nu(r).$$

Combining these two equations and the concentration analysis above, we conclude that with probability at least $1 - \delta$,

$$\langle \theta, \mathcal{U}^*(\theta) \rangle = n \int_\Omega \int_\Omega \theta(w) \theta(r) Q(\mathcal{K}(w, r)) d\nu(w) d\nu(r)$$

$$= n \int_\Omega \int_\Omega \theta(w) \theta(r) \left\{ \frac{Q(\mathcal{K}(w, r))}{Q_n(\mathcal{K}(w, r))} \right\} Q_n(\mathcal{K}(w, r)) d\nu(w) d\nu(r)$$

$$\leq \left( 1 + \frac{48\sqrt{d \log(2L_0 A)} + 2\sqrt{\log(1/\delta)}}{\eta\sqrt{n}} \right) n \int_\Omega \int_\Omega \theta(w) \theta(r) Q_n(\mathcal{K}(w, r)) d\nu(w) d\nu(r)$$

$$= \left( 1 + \frac{48\sqrt{d \log(2L_0 A)} + 2\sqrt{\log(1/\delta)}}{\eta\sqrt{n}} \right) \langle \theta, \mathcal{U}_\mathcal{D}(\theta) \rangle.$$

Replacing the general $\theta$ by $\widehat{\theta}_{\mathcal{D}} - \theta^*$ an using Corollary 3.8, it holds that, with probability at least $1 - 2\delta$,

$$||\widehat{\theta}_{\mathcal{D}} - \theta^*||_{\mathcal{U}^*} \leq \left(1 + \frac{48\sqrt{d\log(2L_0 A)} + 2\sqrt{\log(1/\delta)}}{\eta\sqrt{n}}\right)^{1/2} \mathcal{E}_\delta(n).$$

On the other hand, by directly calculating $||\widehat{\theta}_{\mathcal{D}} - \theta^*||_{\mathcal{U}^*}^2$, we find that

$$
\begin{aligned}
||\widehat{\theta}_{\mathcal{D}} - \theta^*||_{\mathcal{U}^*}^2 &= \left\langle \widehat{\theta}_{\mathcal{D},\varepsilon} - \theta^*, \mathcal{U}^*(\widehat{\theta}_{\mathcal{D}} - \theta^*)\right\rangle \\
&= n\int_{\mathcal{X},\mathcal{A}}\int_S \int_\Omega \left(\widehat{\theta}_{\mathcal{D}} - \theta^*\right)(w)\int_\Omega \left(\widehat{\theta}_{\mathcal{D}} - \theta^*\right)(r)\phi(x,a,w,t)\phi(x,a,r,t)dm(s)d\nu(w)d\nu(r)dQ_{x,a} \\
&= n\int_{\mathcal{X},\mathcal{A}}\int_S \left(\int_\Omega \left(\widehat{\theta}_{\mathcal{D}} - \theta^*\right)(w)\phi(x,a,w,t)d\nu(w)\right)\left(\int_\Omega \left(\widehat{\theta}_{\mathcal{D}} - \theta^*\right)(r)\phi(x,a,r,t)d\nu(r)\right)dm(s)dQ_{x,a} \\
&= n\mathbb{E}_{x,a}\left[||\widehat{F}_{\mathcal{D}}(x,a,s) - F^*(x,a,s)||_{L^2(S,m)}^2\right].
\end{aligned}
$$

Assembling all these parts, we finally get the following inequality,

$$\mathbb{E}_{x,a}\left[||\widehat{F}_{\mathcal{D}}(x,a,s) - F^*(x,a,s)||_{L^2(S,m)}^2\right] \leq \frac{\left(1 + (48\sqrt{d\log(2L_0 A)} + 2\sqrt{\log(1/\delta)})/\eta\right)\mathcal{E}_\delta(n)^2}{n}.$$

By Lemma 3.7, we know that

$$\mathcal{E}_\delta(n) \triangleq 2\log(1/\delta)^{1/2} + \left(2\sqrt{s_0 \log(1+n)} + M\right)n^{\frac{\gamma}{\gamma+2}}.$$

By denoting $C(d, L_0, \delta, A, \eta)$ as $\left(1 + (48\sqrt{d\log(2L_0 A)} + 2\sqrt{\log(1/\delta)})/\eta\right)$, we finish our proof. $\square$

# E. Proofs in Section 4

**Lemma E.1** ((Simchi-Levi & Xu, 2021))**.** *Assume that we are given an offline regression oracle* RegOff *and i.i.d. data* $\mathcal{D} = \{(x_i, a_i, r_i)\}_{i=1}^n$ *where* $\mathbb{E}[r_i|x_i, a_i] = f^*(x_i, a_i)$. *With probability at least* $1 - \delta$, *it returns* $\widehat{f} : \mathcal{X} \times \mathcal{A} \to \mathbb{R}$ *such that*

$$\mathbb{E}_{x,a}\left[\left(\widehat{f}(x,a) - f^*(x,a)\right)^2\right] \leq \frac{\mathsf{Est}_\delta(n)}{n}$$

*for some number* $\mathsf{Est}_\delta(n)$. *Then, define epoch schedule* $\xi_m = 2^m$ *and exploration parameter* $\varsigma_m = \frac{1}{2}\sqrt{K/\mathsf{Est}_{\delta/m^2}(\xi_{m-1})}$. *For any* $t$, *let* $m(t)$ *be the number of epochs that round* $t$ *lies in. For any* $T$ *large enough, with probability at least* $1 - \delta$, *the regret of Algorithm 2 after* $T$ *rounds is at most*

$$\mathcal{O}\left(\sqrt{K}\sum_{m=2}^{m(T)}\sqrt{\frac{\mathsf{Est}_{\delta/(2m^2)}(\xi_{m-1} - \xi_{m-2})}{\xi_{m-1} - \xi_{m-2}}}(\xi_m - \xi_{m-1})\right).$$

*Proof of Theorem 4.2.* From our regression oracle, we have that

$$\frac{\mathsf{Est}_\delta(n)}{n} = \frac{L^2 C(d, L_0, \delta/2, A, \eta)\mathcal{E}_{\delta/2}(n)^2}{n}$$

By applying Lemma E.1 and plugging in the choices that $\xi_m = 2^m$, we get

$\text{Reg}(T)$

$$\leq \mathcal{O}\left(\sqrt{K}\sum_{m=2}^{m(T)}\sqrt{\frac{\text{Est}_{\delta/(2m^2)}(\xi_{m-1}-\xi_{m-2})}{\xi_{m-1}-\xi_{m-2}}}(\xi_m-\xi_{m-1})\right) + \mathcal{O}(1)$$

$$\leq \mathcal{O}\left(\sqrt{K}L\sum_{m=2}^{m(T)}\sqrt{C(d,L_0,\delta/4m^2,A,\eta)}\frac{\mathcal{E}_{\delta/(4m^2)}(\xi_{m-1}-\xi_{m-2})}{\sqrt{\xi_{m-1}-\xi_{m-2}}}(\xi_m-\xi_{m-1})+1\right)$$

$$\leq \mathcal{O}\left(\sqrt{K}L\sum_{m=2}^{m(T)}\sqrt{C(d,L_0,\delta/4m^2,A,\eta)}\frac{2\log(4m^2/\delta)^{1/2}+\left(2\sqrt{s_0\log(1+2^{m-2})}+M\right)(2^{m-2})^{\frac{\gamma}{\gamma+2}}}{2^{m/2-1}}\cdot 2^{m-1}+1\right)$$

$$(8)$$

By our epoch schedule, it holds that $m(T) \leq \lceil\log_2(T)\rceil$. By Theorem 3.10, we know

$$C(d,L_0,\delta/4m^2,A,\eta) = \left(1+(48\sqrt{d\log(2L_0A)}+2\sqrt{\log(4m^2/\delta)})/\eta\right) \leq \frac{192\sqrt{d\log(2L_0A)}}{\eta}\sqrt{2\log(m/\delta)}.$$

Thus,

$$C(d,L_0,\delta/4m^2,A,\eta) \leq \mathcal{O}\left(\frac{\sqrt{d\log(2L_0A)}}{\eta}\sqrt{\log(m/\delta)}\right).$$

Therefore, for Equation (8), we further have

$$\text{Reg}(T) \leq \mathcal{O}\left(\frac{L\sqrt{dK\log(2L_0A)}}{\eta}\sqrt{\log(m/\delta)}\sum_{m=2}^{m(T)}\left((\log(\log T/\delta))^{1/2}+\left(\sqrt{s_0\log(1+\log T)}+M\right)2^{\frac{\gamma(m-2)}{(\gamma+2)}}\right)2^{m/2}\right).$$

Denote the number $\frac{L\sqrt{dK\log(2L_0A)}}{\eta}$ as $\widetilde{C}(K,L,L_0,A,d,\eta)$. We have,

$$\mathcal{O}\left(\frac{L\sqrt{dK\log(2L_0A)}}{\eta}\sqrt{\log(m/\delta)}\sum_{m=2}^{m(T)}\left((\log(\log T/\delta))^{1/2}+\left(\sqrt{s_0\log(1+T)}+M\right)2^{\frac{\gamma(m-2)}{(\gamma+2)}}\right)2^{m/2}\right)$$

$$\leq \widetilde{C}(K,L,L_0,A,d,\eta)\mathcal{O}\left(\log(\log T/\delta)\sum_{m=2}^{m(T)}2^{m/2}+(\log\log T/\delta)^{1/2}\sum_{m=2}^{m(T)}\left(\sqrt{s_0\log(1+\log T)}+M\right)2^{\frac{(3\gamma+2)m}{2(\gamma+2)}}\right)$$

$$\leq \widetilde{C}(K,L,L_0,A,d,\eta)\mathcal{O}\left(\log(\log T/\delta)\sqrt{T}+\left(s_0^{1/2}+M\right)(\log\log T/\delta)T^{\frac{3\gamma+2}{2(\gamma+2)}}\right)$$

$$\leq \widetilde{C}(K,L,L_0,A,d,\eta)\left((s_0^{1/2}+M)\vee 1\right)\mathcal{O}\left(\log(\log T/\delta)\left(\sqrt{T}+T^{\frac{3\gamma+2}{2(\gamma+2)}}\right)\right).$$

$\square$

*Proof of Theorem 4.1.* By the $L$-Lipschitz continuity of $\mathcal{T}$, we have

$$\mathbb{E}_{x,a}\left[\left(\mathcal{T}(\widehat{F}_\mathcal{D}(x,a,s))-\mathcal{T}(F^*(x,a,s))\right)^2\right] \leq \mathbb{E}_{x,a}\left[L^2\|\widehat{F}_\mathcal{D}(x,a,s)-F^*(x,a,s)\|^2_{L^2(S,m)}\right]$$

Then we apply Theorem 3.10 and replace $2\delta$ by $\delta$ in the claim of Theorem 3.10. Thus, with probability $1-\delta$,

$$\mathbb{E}_{(x,a)\sim Q}\left[\|\widehat{F}_\mathcal{D}(x,a,s)-F^*(x,a,s)\|^2_{L^2(S,m)}\right] \leq \frac{C(d,L_0,\delta/2,A,\eta)\mathcal{E}_{\delta/2}(n)^2}{n}.$$

Combining these two inequalities together, we shall get our result. $\square$

# F. Computation

Given the theorems and properties we have established, a natural and fundamental question arises: how can we numerically compute the eigenvalues and eigenfunctions of an infinite-dimensional operator? In fact, within the applied functional analysis community, there is a wide range of methods for numerically solving the eigenvalues and eigenfunctions of integral operators in infinite-dimensional spaces (for example, see Chatelin (2011); Ray & Sahu (2013); Chatelin (1981); Kohn (1972); Ahues et al. (2001); Panigrahi (2017) for reference), including the Galerkin method, the Rayleigh-Ritz method, quadrature approximation methods and so on. In this section, we will briefly introduce one method based on degenerate kernels for calculating the eigenvalues and eigenfunctions as an illustration. Please see Gnaneshwar (2007) for full details.

We assume that $\Omega = [0, 1]$ and an integral operator $C$ is defined as $C[x](s) = \int_0^1 k(s,t)x(t)dt$. The basic idea of Algorithm 3 is to construct a finite-dimensional integral kernel $k_{N_h}$ to approximate $k$ holding the property that the differences between their eigenvalues and eigenfunctions are small enough to be ignored. Therefore, to solve the eigenvalues and eigenfunctions for the kernel $k$, we can instead solve them for $k_{N_h}$, which is finite-dimensional and can be readily transformed into a matrix eigenvalue problem.

We present the pseudo-code in Algorithm 3 and omit the theoretical approximation analysis here. Please refer to Gnaneshwar (2007) for concrete theoretical guarantees.

---

**Algorithm 3** Degenerate Kernel Method

---

**Require:** partition number $n$, kernel $k(s,t)$, degree number $r$ and corresponding Legendre polynomials $L_r(t) = \frac{d^r}{dt^r}(t^2 - 1)^r$.

Partition $[0,1]$ into $n$ intervals, $0 = x_0 < x_1 < x_2 < \cdots < x_{n-1} < x_n = 1$, $I_k = [x_{k-1}, x_k]$, $h = \max |I_k|$. Set $f_k(t) = \frac{1-t}{2}x_{k-1} + \frac{1+k}{2}x_k, -1 \leq t \leq 1$.

Define $\mathcal{P}_r$ as the space of polynomials of degree $\leq r-1$. Denote $N_h = nr$.

Define piecewise polynomial space associated with $\mathcal{P}_r$

$$\mathcal{S}_h^r = \left\{ u : [0,1] \to \mathbb{R} : u|_{I_k} \in \mathcal{P}_r, 1 \leq k \leq n \right\}.$$

Find the Gauss point set (zeros set) of $L_r(t)$ in $[0,1]$, $B_r = \{y_1 \cdots, y_r\}$.

Let $A = \cup_{k=1}^n f_k(B_r) = \{\omega_{i,k} = f_k(y_i) : 1 \in [r], k \in [n]\}$ and $l_i(x)$ be the Lagrange polynomials of degree $r-1$ with respect to $y_1 \cdots, y_r$ such that $l_i(y_j) = \delta_{ij}$.

Define $\rho_{jp}(x) = \begin{cases} l_j(f_p^{-1}(x)) & x \in [x_{p-1}, x_p] \\ 0 & \text{otherwise.} \end{cases}$

Notice $\rho_{jp} \in \mathcal{S}_h^r$ and $\rho_{jp}(\omega_{ik}) = \delta_{ji}\delta_{kp}$, for $i, j \in [r], k, p \in [n]$. Set $t_{(k-1)r+j} = \omega_{jk}$, $z_{(k-1)r+j} = \rho_{jk}$, for $k, p \in [n]$.

Let $\mathcal{S}_h^r \otimes \mathcal{S}_h^r$ be the tensor product space of $\mathcal{S}_h^r$ with dimension $N_h = nr$.

Define degenerate kernel

$$k_{N_h}(s,t) = \sum_{i=1}^{N_h} \sum_{j=1}^{N_h} k(\omega_i, \omega_j) z_i(s) z_j(t),$$

which induces a degenerate kernel operator

$$C_{N_h}(x)(s) = \int_0^1 \sum_{i=1}^{N_h} \sum_{j=1}^{N_h} k(\omega_i, \omega_j) z_i(s) z_j(t) x(t) dt.$$

Solve the eigenvalue problem for the $N_h$-dimensional integral kernel $k_{N_h}$.

---

We now provide a method to solve the eigenvalue problem for the finite-dimensional kernel $k_{N_h}$ for completeness.

**Solve eigenvalue problem for finite-dimensional $k_{N_h}$:** Consider the following formula that

$$\lambda \cdot g(s) = \int_0^1 \sum_{i=1}^{N_h} \sum_{j=1}^{N_h} k(\omega_i, \omega_j) z_i(s) z_j(t) g(t) dt. \tag{9}$$

Here, $\lambda$ stands for the unknown eigenvalue and $g$ is the corresponding eigenfunction. Define $c_i \triangleq \frac{1}{\lambda} \int_0^1 \sum_{j=1}^{N_h} k(\omega_i, \omega_j) z_j(t) g(t) dt$, $i \in [N_h]$, and $\boldsymbol{c} = (c_1, c_2, \cdots, c_{N_h})$ is an unknown vector in $\mathbb{R}^{N_h}$ which we will solve for. Then, we can rewrite Equation (9) as

$$g(s) = \sum_{i=1}^{N_h} c_i z_j(s). \tag{10}$$

Plugging Equation (10) back to the definition of $c_i$, it holds that

$$c_i = \frac{1}{\lambda} \int_0^1 \sum_{j=1}^{N_h} k(\omega_i, \omega_j) z_j(t) \sum_{k=1}^{N_h} c_k z_k(t) dt = \frac{1}{\lambda} \sum_{j=1}^{N_h} \sum_{k=1}^{N_h} c_k k(\omega_i, \omega_j) \int_0^1 z_j(t) z_k(t) dt, \ i \in [N_h], \tag{11}$$

which is equivalent to

$$\lambda \cdot c_i = \int_0^1 \sum_{j=1}^{N_h} k(\omega_i, \omega_j) z_j(t) \sum_{k=1}^{N_h} c_k z_k(t) dt = \sum_{j=1}^{N_h} \sum_{k=1}^{N_h} c_k k(\omega_i, \omega_j) \int_0^1 z_j(t) z_k(t) dt, \ i \in [N_h]. \tag{12}$$

Note that within the implementation of Algorithm 3, both $k(\omega_i, \omega_j)$ and $z_j(t), z_k(t)$ are known to us. Besides, the right hand side of Equation (12) is linear with respect to $\boldsymbol{c}$.

Define $b_{ik} = \sum_{j=1}^{N_h} k(\omega_i, \omega_j) \int_0^1 z_j(t) z_k(t) dt$. Then, Equation (12) is equivalent to

$$\lambda \cdot \begin{pmatrix} c_1 \\ c_2 \\ \vdots \\ c_{N_h} \end{pmatrix} = \begin{pmatrix} b_{11} & b_{12} & \cdots & b_{1N_h} \\ b_{21} & b_{22} & \cdots & b_{2N_h} \\ \vdots & \vdots & \vdots & \vdots \\ b_{N_h 1} & b_{N_h 2} & \cdots & b_{N_h N_h} \end{pmatrix} \cdot \begin{pmatrix} c_1 \\ c_2 \\ \vdots \\ c_{N_h} \end{pmatrix}.$$

Equation (12) finally turns out to be a matrix eigenvalue problem with unknown eigenvector $\boldsymbol{c} = (c_1, c_2 \cdots, c_{N_h})^T$ and eigenvalue $\lambda$. We could solve this matrix eigenvalue problem to acquire $(c_1, c_2 \cdots, c_{N_h})$ and eigenvalue $\lambda$. Eventually, we derive the eigenfunction associated with $\lambda$ for kernel $k_{N_h}$ using Equation (10), i.e., $g(s) = \sum_{i=1}^{N_h} c_i z_i(s)$.

## G. Auxiliary Results

**Definition G.1** (Gohberg et al. (2012))**.** Let $A$ be a compact operator in Hilbert space $H$ and let $\lambda_1(A^*A) \geq \lambda_2(A^*A) \geq \cdots$ be the sequence of non-zero eigenvalues of $A^*A$. $A^*$ is the adjoint operator of $A$. Then the $j$ th singular value of $A$ $s_j(A)$ is defined as

$$s_j(A) \triangleq (\lambda_j(A^*A))^{1/2}.$$

*Property* G.2. For any self-adjoint positive Hilbert-Schmidt integral operator $\mathcal{U}$, it holds that

$$\lambda_j(\mathcal{U}) = s_j(\mathcal{U}) \text{ for } \forall j.$$

*Proof.* We denote the integral kernel of $\mathcal{U}$ as $\mathcal{K}$. Because $\mathcal{U}$ is self-adjoint positive Hilbert-Schmidt integral operator, then for $\forall \theta$,

$$\mathcal{U}(\theta) = \sum_{i=1}^{\infty} \lambda_i(\mathcal{U}) \langle \theta, e_i \rangle e_i.$$

Therefore,

$$\mathcal{U}(\mathcal{U}(\theta)) = \sum_{i=1}^{\infty} \lambda_i^2 \langle \theta, e_i \rangle e_i.$$

Because $\mathcal{U}^* = \mathcal{U}$, we finish the proof. $\square$

**Theorem G.3** (Corollary 3.6 in Gohberg et al. (2012)). *If $A$ and $B$ are compact operators on the Hilbert space $H$, then for any $n$,*

$$\sum_{j=1}^{n} s_j(A+B) \leq \sum_{j=1}^{n} s_j(A) + \sum_{j=1}^{n} s_j(B).$$

**Theorem G.4** (Corollary 4.2 in Gohberg et al. (2012)). *Let $A$ and $B$ be compact operators on a Hilbert space $H$, then for any $p > 0$ and any $k$,*

$$\sum_{j=1}^{k} s_j^p(AB) \leq \sum_{j=1}^{k} s_j^p(A) s_j^p(B),$$

*and*

$$\sum_{j=1}^{k} s_j(AB) \leq \sum_{j=1}^{k} s_j(A) s_j(B).$$

Now we introduce the functional determinant of a trace class operator. We notice here that the formal definition of functional determinant requires exterior product and tensorization of the Hilbert space $H$. Nonetheless, by Lidskii's Theorem (Simon, 2005), we are able to achieve an equivalent characterization of it, so we directly use this characterization as a definition for simplicity.

**Definition G.5** (Theorem 3.4.7 in Kostenko). For any trace class operator $A$, the functional determinant $\det(I + zA)$ is defined as

$$\det(I + zA) \triangleq \prod_i (1 + z\lambda_i(A)), \ \forall z \in \mathbb{C}.$$

**Theorem G.6** (Theorem 3.10 in Simon (2005)). *Suppose the integral operator $A$ is defined as $Af(x) = \int_\Omega K(x, y) f(y) dy$, where $\Omega$ is compact and $K(x, y)$ is continuous on $\Omega \times \Omega$. Then, it holds that*

$$tr(A) = \int_\Omega K(x, x) dx.$$

*In the meanwhile,*

$$\det(I + A) = \sum_{i=0}^{\infty} \frac{\alpha_m}{m!},$$

*where*

$$\alpha_m = \int_\Omega \int_\Omega \cdots \int_\Omega \det[(K(x_i, x_j))_{1 \leq i,j \leq m}] dx_1 dx_2 \cdots dx_m, \ \forall \ m \geq 1.$$

*We conventionally set $\alpha_0 = 1$.*

Now we are ready to analyze the integral operator $\mathcal{L}^{x,a}$ and $\mathcal{U}_\mathcal{D}$ in our paper.

**Lemma G.7.** *For any $x, a$, $\det(1 + \mathcal{L}^{x,a}) \leq e$.*

*Proof of Lemma G.7.* We apply Theorem G.6 to the operator $\mathcal{L}^{x,a}$. The associated integral kernel is $\mathcal{K}^{x,a}(w, r) = \int_S \phi(x, a, w, t) \phi(x, a, r, t) dm(t)$. Therefore, by Theorem G.6, we have

$$(K(w_i, w_j))_{i,j=1}^m$$
$$= \begin{pmatrix} \int \phi(x,a,w_1,t)^2 dt & \int \phi(x,a,w_1,t)\phi(x,a,w_2,t)dt & \cdots & \int \phi(x,a,w_1,t)\phi(x,a,w_m,t)dt \\ \int \phi(x,a,w_2,t)\phi(x,a,w_1,t)dt & \int \phi(x,a,w_2,t)^2 dt & \cdots & \int \phi(x,a,w_2,t)\phi(x,a,w_m,t)dt \\ \vdots & \vdots & \vdots & \vdots \\ \int \phi(x,a,w_m,t)\phi(x,a,w_1,t)dt & \int \phi(x,a,w_m,t)\phi(x,a,w_2,t)dt & \cdots & \int \phi(x,a,w_m,t)^2 dt \end{pmatrix}$$

This is a symmetric matrix and

$$\sum_{i=1}^m \lambda_i((K(w_i, w_j))_{i,j=1}^m) = \sum_{i=1}^m \int_S \phi(x, a, w_i, t)^2 dm(t) \leq m,$$

due to $0 \leq \phi \leq 1$ and $m(S) = 1$. Therefore, by the AM-GM inequality,

$$\left(\prod_{i=1}^{m} \lambda_i((K(w_i, w_j))_{i,j=1}^{m})\right)^{1/m} \leq \frac{\sum_{i=1}^{m} \lambda_i((K(w_i, w_j))_{i,j=1}^{m})}{m} \leq 1.$$

We find that

$$\det[(K(w_i, w_j))_{i,j=1}^{m}] = \prod_{i=1}^{m} \lambda_i((K(w_i, w_j))_{i,j=1}^{m}) \leq 1.$$

Recalling $\nu(\Omega) = 1$, we have that for $\forall\, m \geq 1,\ \alpha_m \leq \int_\Omega \int_\Omega \cdots \int_\Omega 1 d\nu(w_1) \cdots d\nu(m) \leq 1$. Thus,

$$\det(1 + \mathcal{L}^{x,a}) = \sum_{i=0}^{\infty} \frac{\alpha_m}{m!} \leq \sum_{i=0}^{\infty} \frac{1}{m!} \leq e.$$

$\square$

