# OpenReview forum: "Contextual Online Decision Making with Infinite-Dimensional Functional Regression"
_ICML.cc/2025/Conference — ICML 2025 poster_

### Official Review · Reviewer_Zz7E · 2025-03-08

**Overall Recommendation:** 2

**Summary:**

This paper consider a contextual decision making problem where the context space is infinite but the decision set is finite. Such kind of formulation applies ,for example, in the contextual Multi-Armed Bandit model. The authors focuses on learning infinite CDF functions, that define the distribution over the decision making result. However, they do not consider the general case but rather have assumptions regarding the CDFs e.g., Lipshitz and $\gamma$-eigendecay.  Beside the linear case, the authors do not mentioned standard distributions for them such assumption applies. The authors define a function-approximation oracle over the CDF function in class, and use it in a iterative-batch-based algorithm that looks identical to the one of Simchi-Levi and Xu 2020 for contextual bandits. They obtain a regret bound that depends on $\gamma$.

## update after rebuttal
After the rebuttal, although my concerns were partially resulved, I am leaning towrds rejection of this paper for the following reasons:

1. The writing requires a significet improvents; this issue seems to be raised by most of the reviewrs.

2. I agree with the concerns raised by reviewer dr26 regarding assumptions 2.1 and 2.10. They should be justified.

3. In my opinition, the results of this paper are limited to only a small set of function classes that can satiafy those assumptions and  the eigen-decay condition. This was also raised by reviewer dr26.

4. The algorithm is an application of the well known Inverse Gap Weigthening (IGW) technique to the presented setting, hence the algorithmic novelty  is limited.

**Claims And Evidence:**

Yes.

**Essential References Not Discussed:**

Significant line of contextual RL literature is not discussed in the main paper: contextual MDPs, most of the works in contextual MAB (that has been vastly studied).

**Experimental Designs Or Analyses:**

The authors present "utility-regret" bound.
I read the text in the main paper - seems sound.

**Methods And Evaluation Criteria:**

Theoretical analysis of regret.

**Other Comments Or Suggestions:**

See questions and Weakness.

**Other Strengths And Weaknesses:**

Strengths:
Interesting research question.

Weakness:
1.The writing requires improvement: the order of things makes it hard to follow, very technical paper with significant lack of intuition behind it and insignificant comparison to related work.
2 The actions comes from a finite set? How this result is different from other papers in CMAB literature that consider infinite context space? Why do not extend the result to infinite action space as well?
3.See questions to the authors.

**Questions For Authors:**

1. What is " oracle inequality" ? You mean oracle regret?

2. Is the proposed oracle efficient? Highlight  the difference of it from the standard ERM oracle. Why this oracle is used instead of standard ERM?

3. As far as I understand your approach, you established some kind of confidence bound around the distribution approximation. Is that correct?

4. Foster and Rakhlin (2020) also considered in the second part of their work and infinite context space. Please compare yourself to their results.

5. Since you assume both boundness and Lipchitzsness, the contribution is unclear to me. There are works (e.g., Modi et al. (2018)) that considers such settings. Please state your contribution clearly.

6. What is the difference between "utility-regret" and the standard contextual pseudo regret?

7. From the algorithm it seems that the actins space is finite, and only the context space is infinite. How does this improve over existing literature? For example, the work for Simchi-Levi and Xu (2021) can be easily extended to infinite context space using a dimension that capture the context-space complexity. Since you already have boundnesses and Lipschitzness assumption, the contribution is unclear to me given previous literature that can be applies to the more general case of infinite context space.

8. Have you tried to consider also infinite action space? It seems the work of Zhu et al. (2022) cover both for bi-linear contextual bandits.

[1] Beyond UCB: Optimal and Efficient Contextual Bandits with Regression Oracles, Foster and Rakhlin (2020).

[2] Bypassing the Monster: A Faster and Simpler Optimal Algorithm for Contextual Bandits under Realizability, Simchi-Levi and Xu (2021).

[3] Contextual Bandits with Large Action Spaces: Made Practical, Zhu et al. (2022)

[4] Markov Decision Processes with Continuous Side Information, Modi et al. (2018)

**Relation To Broader Scientific Literature:**

I do not see immediate implications beyond contextual MAB.

**Theoretical Claims:**

Read the text in the main paper - seems sound.

---

> ### Author Rebuttal · Authors · 2025-04-01
>
> Thank you for your questions. We answer your questions in order.
>
> **RE:oracle ineq** **Oracle inequality** provides an upper bound on the performance of the learning algorithm compared to an ideal benchmark ("oracle"), the best function in a reference class. A typical form is:
>
> $L(\hat{f}) \leq L(f^*) + \text{complexity penalty},$
>
> where $\hat{f}$ is the output of algorithm, $f^*$ is the best-in-class function, and $L(\cdot)$ denotes the loss. The additional term represents the complexity penalty.
>
> **Oracle Regret.**
> In contrast, **oracle regret** is a concept from online learning. It measures the cumulative performance gap between the learner and the best fixed decision in hindsight:
>
> $\text{Reg}(T) = \sum_{t=1}^{T} L_t(f_t) - \min_{f \in F} \sum_{t=1}^{T} L_t(f)$
>
> where $f_t$ is the learner's decision, and $L_t$ is the loss function at time $t$. Oracle regret measures dynamic performance; sublinear regret means average vanishes as $T \to \infty$.
>
> Therefore, oracle regret is not oracle inequality.
>
> **RE: oracle efficiency** Our adaptive algorithm solves uncountable-dimension functional optimization problems efficiently by adaptive approximation with good statistical guarantee.
> ERM approach is intractable because it is practically impossible to optimize an infinite-dimensional functional optimization problem.
>
> **RE: confidence bound** For confidence bound, please refer to Thm 3.6. This constructs a function confidence bound. **For arxiv.org/abs/2002.04926**, they assume access to abstract online oracle and is neither practical nor valid in our problem. Also, it only cares about mean reward setting, which is just a subclass of problems that we can handle.
>
> **RE: Boundness and Lipschitzness** are both common assumptions in online learning. A lot of function classes are Lipschitz. All linear functions are Lipschitz;  neural networks are Lipschitz (www.mit.edu/~rakhlin/courses/mathstat/rakhlin_mathstat_sp22.pdf). Our Lipschitz condition concerns parameter distance $||w - r||$ in the family.
>
> $|\phi(x,a,r,s)-\phi(x,a,w,s)|\le L_0||w-r||_{\infty}.$
>
> This assumption could be found in other papers, see arxiv.org/pdf/2007.07876, proceedings.mlr.press/v15/chu11a/chu11a.pdf.
>
> The boundness assumption is common in online learning; see https://proceedings.neurips.cc/paper/2011/file/e1d5be1c7f2f456670de3d53c7b54f4a-Paper.pdf , arxiv.org/abs/1611.06426, arxiv.org/pdf/2106.03365, https://openreview.net/forum?id=F5TbbyTgbC.
> In finite dimensions, it assumes $\theta^*\in R^d$, $||\theta^*||_2\le S$.
>
> Our contribution is not about ** relaxing these conditions.** Instead, we propose a
> **framework for contextual online decision-making** capable of a wide range of tasks, including bandits, online hypothesis testing, and risk-aware bandits. We design **efficient regression oracle for infinite-dimensional functional regression**. By spectral decomposition and eigenvalue truncation, we solve an infinite-dimensional function optimization problem efficiently by adaptive approximation. Combined with inverse-gap weighting, this oracle yields our algorithm.
>
> Our theoretical contribution is that we provide **characterization of the
> relationship between regret and the eigendecay rate of the operator** by single parameter $\gamma$. This is the first regret bound for infinite-dimensional decision-making via eigendecay.
>
> **RE: Pseudo regret** Pseudo-regret is trajectory-based and random; utility regret is its expected version which is common in literature.
>
> **RE: context and action space** Our paper could handle arbitrary context space. We study finite action space purely for simplicity. arxiv.org/abs/2207.05836 can handle infinite action space because its reward model has a special structure.
>
> To extend to infinite action space,  we change the algorithm to a UCB-type algorithm to handle it, see arxiv.org/pdf/2007.07876. We define the following action divergence $V_x(a||\lbrace a_i\rbrace_{i=1}^{n})$ such that
>
> $V_x(a||\lbrace a_i\rbrace_{i=1}^{n})\ge \sup_{\theta}\Big\lbrace\frac{|T(\langle\theta(\cdot),\phi(x,a,\cdot,\cdot)\rangle)-T(\langle\theta^*(\cdot), \phi(x,a,\cdot,\cdot)\rangle)|^2}{\sum_{i=1}^{n}(T(\langle\theta(\cdot), \phi(x,a_i,\cdot,\cdot)\rangle)-T(\langle\theta^*(\cdot), \phi(x,a_i,\cdot,\cdot)\rangle))^2}\Big\rbrace.$
>
> This divergence is a UCB.
> At any round $t$, receiving context $x_t$, for $i=d_x+1,\cdots,t$, where $d_x$ is some parameter, we pretend all previous actions were applied at context $x_t$ and simulate the counterfactual action sequence by solving
>
> $a_{ti} \in argmax_{a \in A} T(\langle \hat\theta_i,\phi(x_t,a)\rangle) + \beta_i V_{x_t}(a || [a_{tj}]_{j=1}^{i-1})$.
>
> Then we apply $a_{tt}$. This algorithm can handle infinite action space with counterfactual action divergence. We will add this discussion to the camera-ready version.
> We believe we have addressed your concerns and hope that you will consider raising the score accordingly.

---

### Official Review · Reviewer_dr26 · 2025-03-08

**Overall Recommendation:** 2

**Summary:**

This paper proposes a general framework for contextual online decision-making problems. The unique challenge about the general setting lies in the estimation the ground-truth distribution $F^*$ which is a function itself, and learning the distribution becomes an infinite-dimensional functional regression problem. Under some assumptions, this paper reduces this problem to learning the leading basis CDFs from the decomposition of the ground-truth. An algorithm is proposed to adaptive collect data and make decisions on-the-fly, with an online estimation of the coefficient $\theta$ given by the regression sub-routine.

## update after rebuttal

I appreciate the authors' response. My original concerns were partially resolved and I would increase the score. However, I still lean towards the reject side for the following reasons:

(1) the justification of assumptions (i.e. basis family $\Phi$) is not enough: from a purely theoretical perspective, it might be okay to omit details on $\Phi$ despite that it might be impractical to implement. Your work is claimed to be a general framework that is capable of subsuming downstream practical applications (examples 2.4-2.6), to support this claim it's necessary to show a reasonable parameterization and implementation of $\Phi$ tailored to each of the examples. Otherwise there will be a gap between your theory and application. A related issue is the lack of details on the examples. Concrete details on how these examples are written within your framework in a practical way should be included (at least in the appendix), your last response only included one example and it still lacks details on assumption 2.1.

(2) the eigen-decay condition should be emphasized: I do appreciate your contribution and I didn't say the eigen-decay condition is not valid. However, it's a small subset of the whole infinite-dimensional functional regression problem class, and exhibits nice "finite-effective-dimension" like property that mitigates the core difficulty of infinite dimension. The current writing is a bit over-selling in my opinion, and I recommend reflecting the eigen-decay condition in your title.

In all, I believe the paper can be greatly improved with a thorough re-writing.

**Claims And Evidence:**

No. I'm mainly concerned about the setting and assumptions. I feel more justification is required.

(1) in assumption 2.1, it's assumed that we have access to a family of basis CDFs, whose (convex) span is guaranteed to contain the ground-truth $F^*$. How do we find such a family without the knowledge of $F^*$? And what kind of oracle access do we have to the family? I feel the family has to be explicitly parameterized by $w$, otherwise for a family with uncountable size, what is a reasonable way to store and access it? The general framework proposed is claimed to be able to subsume previously considered settings (examples 2.4-2.6). However, I feel the examples are not explained in enough details: i.e. what's the basis family $\Phi$ in each of the example? And do they satisfy assumptions 2.1-2.3?

(2) in assumption 2.10, it's assumed that the eigenvalues are decaying fast enough, dominated by a sequence $\tau_k$. In my opinion, this is essentially a finite effective dimension assumption, which disenchants the "magic" of the infinite-dimensional functional regression storyline, which is the essential contribution of your work. The justification paragraph before assumption 2.10 is farfetched and not convincing (e.g. "In the analysis of many machine learning algorithms"). How are the listed references related to this work? Can you provide more direct justifications? In addition, can you explain more on the sequence $\tau_k$ and why the two constraints make sense? For example, the constraint $\tau_k=O(\frac{1}{k})$ seems loose, because $\tau_k=\Theta(\frac{1}{k})$ will contradict the first constraint.

In all, the technical setting and assumptions should not be casual. I would appreciate more justifications.

**Essential References Not Discussed:**

No.

**Experimental Designs Or Analyses:**

NA

**Methods And Evaluation Criteria:**

Yes.

**Other Comments Or Suggestions:**

NA.

**Other Strengths And Weaknesses:**

I feel the writing is not clear enough: some formulas and math are not necessary for the main-text, please considering moving them to the appendix for more explanatory writing in the main-text.

The paper's organization is "straight", it lists the components one by one, and I get lost between the transitions of these components in the first pass of reading. Please consider adding a concise section explaining the main idea and technical roadmap before section 2.1.

**Questions For Authors:**

See 'Claims And Evidence".

**Relation To Broader Scientific Literature:**

It's related to both machine learning and operation research areas.

**Theoretical Claims:**

No, I'm not convinced by the problem setting and assumptions in the first place.

---

> ### Author Rebuttal · Authors · 2025-04-01
>
> Thank you for your kind questions, and we would now like to answer your questions in order.
>
> **RE: Model Assumption 2.1** Thank you for your question. In machine learning, it is a common assumption to assume that the underlying true model lies in some known model class. This assumption is usually called the realizability assumption, and we follow this protocol. See arxiv.org/abs/2107.02237, arxiv.org/abs/2410.12713 ,arxiv.org/abs/2002.04926 . Thus, we assume that there is a basis CDF family
> $\Phi= \lbrace\phi(x,a,w,s)\rbrace_{w\in\Omega}$.
> The true distribution function is of the form $F^*(x,a,s)=\int_{\Omega}\theta^*(w)\phi(x,a,w,s)d\nu(w)$. As for the estimation oracle, this is our key contribution. Generally, estimating the true $\theta^*(w)$ is an infinite-dimensional functional optimization problem and is impossible to solve. However, by spectral decomposition in functional analysis and eigenvalue decomposition, we successfully developed an efficient estimation oracle.
>
> With no prior knowledge about the model class, you can still estimate using any model class.
> In this circumstance, there will be two types of errors, the first type is **approximation
> error** induced by potential model misspecification. This is unavoidable because the assumption about the underlying true model is wrong. The second type is estimation error, which can be reduced by designing more delicate algorithms. In this paper, we assume a well-specified model class and focus on reducing the estimation error. In practice, considering storage space and computation, we can choose a moderate neural network class as model candidates. In this case, we will suffer from approximation error, and our paper provides a theoretical bound on the estimation error. By balancing computation and performance—where a larger network reduces approximation error but requires more resources—our work offers theoretical insights for practical applications.
> }
> For the storage and of such a distribution family, we can first view $\Phi$ just as a function of $x,a,w,s$. We could use a neural network to store and learn such nonlinear functions. For example, see arxiv.org/abs/2110.03177, arxiv.org/abs/2306.00242 , arxiv.org/abs/2305.03784 .
>
> **RE: Basis Family $\Phi$** Modeling the basis function family could be problem-driven, different tasks have different families. Generally, we use spline functions, trigonometric functions, truncated Gaussian mixtures, and the Bernoulli random variable mixture to model the basis distribution family in different applications. Please see arxiv.org/abs/2205.14545 for numerical details.
>
> **RE: Assumption 2.10**
> Thanks for your question. First, the eigendecay rate assumption is not a finite effective dimension assumption. The infinite dimension setting actually makes the traditional regret bound of online learning invalid. For example, in linear bandit papers with dimension $d$, often the optimal regret rate after $T$ rounds is $\mathcal{O}(\sqrt{dT})$,
> and if we let $d$ go to $\infty$, we incur an invalid regret bound.
>
> Assumption 2.10 is a polynomial eigendecay assumption, which is common in many machine learning subfields such as kernel learning, deep neural network analysis arxiv.org/abs/2305.02657, and neural network learnability analysis arxiv.org/abs/1708.03708. Some even assumed an exponential eigendecay rate. We use a polynomial eigendecay rate to characterize the estimation error of our functional regression oracle. Our assumption is mild in comparison.
>
> Regarding the question about $O(1/k)$ and $\Theta(1/k)$, we give a counterexample that a finite sum doesn't imply $O(1/k)$. For example, the series $a_i=\frac{1}{i^{2}}$ for $i\neq 2^{2j},\ j=1,2,\cdots$, and $a_i=\frac{1}{\sqrt{i}},\ i=2^{2j},\ j=1,2,\cdots$.
> Also, please see Prop 2.9 before jumping into Assumption 2.10. In Prop 2.9, we first prove that the sum of the eigenvalues is finite. Mathematically, this is called trace-class, so it is impossible that $\lambda_k=\Theta(1/k)$ as the sum would be infinite, which violates Prop 2.9.
>
> The first line of 2.10 is that decay rate of sequence $\{\tau_k\}$ is strictly faster than $1/k$. For the second line, the first inequality says that the eigenvalue sequence of the operator could be dominated by $\{\tau_k\}$, which cannot be induced by the first line. The second inequality is a restatement of the third point in Prop 2.9, saying that the eigenvalues are dominated by the $\{\frac{1}{k}\}$.
>
> **RE: Technical Roadmap**
> We will write a clear roadmap in the camera-ready version and improve the structure of our paper.
>
> Please feel free to discuss with us any further questions. We hope our rebuttal has clarified the reviewer’s confusion and respectfully hope that the reviewer would consider re-evaluating the merit of our work accordingly.

---

> > ### Comment · Reviewer_dr26 · 2025-04-02
> >
> > Thank you for your reply. In general, I feel it's not very useful to answer my instance-specific questions by reasoning that some condition is common in a more general problem. They are usually not close enough and I would appreciate details tailored to your specific problem setting. Please avoid using terms like "in machine learning" which is too abstract.
> >
> > **RE: Model Assumption 2.1** Sure in general machine learning realizability is a common assumption. But an assumption being common doesn't necessarily mean it's universally legit: it may be widely applicable because it's mild in many natural settings, but a common assumption may also have scenarios that it doesn't make sense.
> >
> > For your specific problem setting, I can restate my questions to perhaps make them clearer: (1) in what scenario, the "size" of $\Phi$ is much smaller than the "size" of all functions in $\mathbf{R}^d$? Say we work with Gaussian measure in $L^2$ space, then as long as $\Phi$ has a countable size, its convex span should have zero measure (correct me if I'm wrong), meaning it's highly inexpressive. Then your assumption that $F^*$ lies in the convex span means either you have very strong prior knowledge on $F^*$ which makes your contribution not as general, or $\Phi$ has an uncountable size which raises my second question on oracle: (2) by oracle I mean the oracle access to the function class $\Phi$, not the estimation oracle. When $\Phi$ has an uncountable size, if it's not parameterized (like family of polynomials), what might be a practical way to access it?
> >
> > You didn't answer my questions on how your framework subsumes examples 2.4-2.6 (for each of them, fit it into your framework and prove it satisfies all your assumptions).
> >
> > **RE: Assumption 2.10** what I mean was the decaying eigenvalue assumption is very alike to a low (constant)-rank assumption in finite (but high) dimensional space problems. As an example, for a PSD matrix with ambient dimension say 1 billion, if its trace is 10, the "effective dimension" trace is more informative than the ambient dimension. Here in your problem when we assume the sum of all eigenvalues is bounded by a constant, it drastically simplifies the problem because essentially we don't need to care about higher-order terms after a small (constant, or maybe logarithmic) number of eigenvectors. Then it effectively reduces the infinite-dimensional problem to a finite-dimensional one right? Then your contribution looks to me should be more accurately described as identifying a sub-class of all infinite-dimensional problems which is essentially finite-dimensional (and we already know how to solve it), instead of proposing a general framework for all infinite-dimensional problems.
> >
> > Thank you for the discussion on the two conditions of $\tau_k$.

---

> > > ### Author Response · Authors · 2025-04-02
> > >
> > > Thank you for your comment. First, the candidate distribution family in our assumption is pamatrized by $w$, and $w\in\Omega\subset R^d$, so $\Phi$ has uncountable size and it is parametrized. Polynomial is just one way. By Stone-Weistrass theorem, we could use polynomial to uniformly approximate any continuous functions on an interval so we believe it is an effective way to parametrize the distribution class.
> > >
> > > For how our framework subsumes examples. We would like to use mean as an example. To prove the Lipschitz continuity, $T(F)=\int_{S}sdF(s)$,
> > >
> > > $|T(F_1)-T(F_2)|=|\int_{S}sd(F_1(s)-F_2(s))|,$
> > >
> > > By integrating by parts and noticing that $S=[a,b]$, $F_1(a)=F_2(a)=0$ and $F_1(b)=F_2(b)=1$, we have,
> > >
> > > $|\int_{S}sd(F_1(s)-F_2(s))|=|\int_{S}F_1(s)-F_2(s)ds|\le \int_{S}|F_1(s)-F_2(s)|ds$
> > >
> > > By Cauchy-Schwarz Inequality, we have
> > >
> > > $\int_{S}|F_1(s)-F_2(s)|ds\le m(S)^{1/2}||F_1-F_2||_{L^2(S)}$
> > >
> > > So the expectation functional is Lipschitz continuous with respect to our norm, for MV bandits and online hypothesis testing we could do similar computation. So, combined with the fact that we can parametrize an uncountable size distribution family, our proposed framework subsumes these examples
> > >
> > > Thank you for your eigenvalue question. For the eigenvalue decay question, your claim that the finite sum of eigenvalue sequence simplifies the question is not true. In fact, any trace-class operator has finite eigenvalue sums, but there is no unified method to abstractly learn any trace-class operator in a Banach space without full information feedback.  Actually, in operator theory and functional analysis, there is a rigorous description about when an integral operator is trace-class, see for example https://www.jstor.org/stable/2047610?seq=1
> > >
> > > If you only use the property that the sum of eigenvalue sequence is finite without eigendecay depict, you would need to add a regularizer which scales with the sample size to do distribution functional regression, which is not applicable in our setting. Please refer to  Functional linear regression of cumulative distribution functions by Zhang et al (2022). The bound in that paper is not valid because the scaling regularizer makes the error bound too large.
> > >
> > > Therefore,  we investigate deeper into that and discover that the eigendecay rate is much more powerful in depicting the estimation error and we propose an efficient adaptive approximation method to solve that.
> > >
> > > Moreover, for your intuition, after a small (constant, or maybe logarithmic) number of eigenvectors, the eigenvalues are much smaller, it essentially means that the eigenvalues are decaying exponentially, which is a stronger assumption than polynomial decay condition in our paper, as for example, if $\gamma=1/2,$ then $\lambda_n\le \frac{1}{n^2}\le\epsilon$ leads to $n>\frac{1}{\epsilon^2}$, this is polynomial with respect to any threshold.

---

### Official Review · Reviewer_aBVe · 2025-03-13

**Overall Recommendation:** 3

**Summary:**

The paper proposes a unified framework for contextual online decision-making using infinite-dimensional functional regression. It introduces an efficient regression oracle to estimate context-dependent CDFs, enabling sublinear regret across tasks. The authors establish a regret bound linked to the eigendecay rate and design an inverse gap weighting policy for efficient learning.

**Claims And Evidence:**

Yes

**Essential References Not Discussed:**

No

**Experimental Designs Or Analyses:**

Yes

**Methods And Evaluation Criteria:**

Yes

**Other Comments Or Suggestions:**

No.

**Other Strengths And Weaknesses:**

One weakness is in the minimax upper bound, since \gamma>0, there is still a gap between their upper bound and \sqrt{T} lower bound.

**Questions For Authors:**

1. What is the novelty of your algorithm?

**Relation To Broader Scientific Literature:**

including contextual bandits, functional regression, operator learning, and decision-making under uncertainty

**Theoretical Claims:**

Yes

---

> ### Author Rebuttal · Authors · 2025-04-01
>
> Thank you for your kind remarks and questions, and we would now like to answer your questions in order.
>
> **RE: Minimax bound**
> Thank you for your question. In the conclusion part, we actually point out that investigating the minimax lower bound of the regret with respect to the eigendecay rate is an important open question. Historically, people focus on finite-dimension cases and avoid discussing infinite-dimension cases. Our work directly extends the existing finite-dimensional problem to infinite dimension and we believe the eigendecay rate is an efficient tool to describe the learnability of infinite-dimensional models. Our conjecture is that we might be optimal. Nonetheless, it is out of the scope of this paper and is an important future direction.
>
> **RE: Our new contribution**
> Our contribution lies in the following perspectives. We propose a unified
> framework for contextual online decision-making capable of addressing a wide range of tasks, including contextual bandits, online hypothesis testing, and risk-aware bandits. We design an efficient regression oracle for infinite-dimensional functional regression. By applying spectral decomposition and eigenvalue truncation, we solve an infinite-dimensional function optimization problem efficiently with adaptive approximation.
> Combining this functional regression oracle with inverse-gap weighting policy from contextual bandits, we can design our efficient sequential decision-making algorithm.
>
> The key theoretical contribution of our paper is that we provide a rigorous characterization of the
> relationship between regret and the eigendecay rate of the operator, depicted by a single parameter $\gamma$. This is, to the best of our knowledge, the first result that characterizes the regret of infinite-dimensional general sequential decision-making by the eigendecay rate of the operator.
>
> Thank you again for your comments. We hope you find the response satisfactory. Please let us know if you have further questions.

---

### Official Review · Reviewer_t1kP · 2025-03-14

**Overall Recommendation:** 4

**Summary:**

This paper studies the contextual bandits setting, in which it develops a novel method --- with broad applicability --- for making decisions whose utility is allowed to be any (square-integrable) function of entire (contextual) distributions associated to each arm/action. This flexible definition of rewards is in contrast to much of the literature, in which rewards are typically assumed to be a simple function of the arms' distributions, such as means, mean-variance etc.

In a nutshell, the proposed approach consists of two parts: (in each period,) the arms' distributions are estimated via an infinite-dimensional functional regression, followed by estimation of the best decisions to take based on the estimated distributions. The functional regression part crucially utilizes the following technique from operator theory: results/assumptions on the spectrum of the design integral operator are made, and the estimation is regularized by cutting off all terms in the eigendecomposition from some term onwards. The policy estimation part is more standard and utilizes an inverse gap weighting approach. The general bounds are then derived for the performance of both above described parts, and are stated as a function of the properties of the design integral operator's spectrum.

#######
Update after rebuttal:

I have read the authors' response, which does mostly address my questions. I will keep my original score, as I still believe the paper is a thought-provoking and reasonably technically sophisticated contribution to the literature --- but it, however, suffers from issues such as its very suboptimal writing, as I have pointed out in this review and on which there seems to be a broad consensus among the other reviewers. I have also read the discussion with the other reviewers, and I believe the writing of this paper may have led to the ensuing lack of clarity over key assumptions of the paper, i.a. the eigendecay --- and as such, it is on the authors to better motivate and contextualize this assumption in future revisions, in particular making sure readers understand why/when this assumption can be nontrivial.

**Claims And Evidence:**

Yes, as the paper is theoretical in nature and all their claims are substantiated by (to the best of my checking) correct proofs.

**Essential References Not Discussed:**

The literature review is currently fairly sufficient for providing the necessary context for the contribution, but see below for a list of several works that I would like to see added discussion on.

**Experimental Designs Or Analyses:**

N/A --- the paper is theoretical in nature.

**Methods And Evaluation Criteria:**

N/A --- The proposed algorithm for contextual bandits, as well as the associated performance bounds, make sense for the setting. As a theory paper, there are no associated experiments or evaluation criteria.

**Other Comments Or Suggestions:**

See above for presentational suggestions below. As for further feedback, I think that while the literature review is overall sufficient, but of particular interest would be:

(1) an expanded discussion of the Zhang et al (2022) reference --- as far as managed to familiarize myself with it, it offers some infinite-dimensional regression guarantees, but the scaling is with the number of points and the assumptions differ. While this is briefly mentioned in the paper at the appropriate point, as well as in the appendix, but I would appreciate further details on that and on how the proposed new procedure manages to improve on that.

(2) A deep paper in the online learning literature (not strictly the considered "contextual bandits" setting) by Foster et al (2018) (https://arxiv.org/pdf/1803.07617) appears to offer a useful complementary angle: how much estimation (i.e. what sufficient statistics) is actually required to achieve certain guarantees. In another angle on the estimation problem, another paper (https://openreview.net/pdf?id=tyqL1bPl0L), in a full feedback online learning setting, studies accurate/calibrated estimation of general functionals of distributions beyond mean-rewards but subject to the functionals being "elicitable". By contrast to these papers, the approach in the present manuscript is to just estimate the entire distribution and take the functional of the estimated CDF when that's possible. Discussing the difference in approaches and techniques to these references, as well as what the proposed infinite-dimensional approach could imply for these other settings, would, I think, meaningfully contribute to positioning the work in-context and pre-view its possible future applications.

**Other Strengths And Weaknesses:**

1. A major strength of the work, as I alluded to above, is the generality of its proposed methodology and bounds. Through a streamlined infinite dimensional regression-based approach, it is able to capture rewards that can be diverse functionals of the arms' contextual distributions, rather than just e.g. arms' means.

2. In addition, the theory developed in this paper is mathematically nontrivial and previously not particularly explored in the present setting. Therefore, the paper makes a sophisticated, and useful, methodological contribution to the area, starting from the ability to perform infinite-dimensional regression when that is required, and not least including the discovered connection relating regret bounds with spectral properties of setting-specific design operators.

---------

For the weaknesses, I can mainly point out that I am not optimally happy with the clarity of the work's presentation. For such subject matter of potentially big relevance to the community, in my opinion the details of the approach are described in a fairly linear and dry fashion, which could be improved. In particular, certain technical details of import could be insightfully highlighted, including but not limited to:

(a) What is the minimal restriction on the utility functional? If square-integrability were not assumed, would the entire approach break down or is there hope with weaker assumptions?

(b) How is the dependence on \gamma propagated from the regression guarantees to the regret guarantees? In other words, I'd like for an intuitive sketch of the regret dependence on gamma to be provided in the main part, rather than buried in the derivation in the appendix.

(c) Several specific instantiations of the setting are presented early on, but none of them are "worked out" after the algorithm is presented --- meanwhile, it would really help the readers to highlight how the assumptions made on the decision mapping and on the distributions play out in specific settings such as mean, MV bandits and sequential hypothesis testing.

(d) Notation is clunky at several points. E.g. one annoying bit is how the utility is often described as a functional of F(x, a, s) --- s here is confusing, instead much less controversial would be to e.g. index F by a, x as F_{x, a} and then simply write e.g. \Tau(F_{x,a})

**Questions For Authors:**

My main gripes are with the presentation of the paper, so I would appreciate if the authors could address in their response my points above by providing some text/discussion of these points that could later be inserted in the manuscript. (My current positive evaluation of the paper already presumes that these will be addressed in the eventual camera-ready.)

**Relation To Broader Scientific Literature:**

This paper can be viewed as a substantial generalization of many results in the literature on contextual online decision making. The main setting studied in the literature is that of mean-rewards (corresponding to vanilla contextual bandits), and there are also many more specialized settings that study risk-aware rewards (which allow for dependence on risk measures such as variance, quantiles, spectral measures etc.) as well as settings such as sequential hypothesis testing --- but due to various statistical and computational intractability concerns, most of the above settings typically don't permit the reward to be a general functional on the space of arms' distributions.

This paper, by contrast, makes rather generic assumptions on the distributions (such as that there is a functional basis with respect to which the distributional estimation is allowed to proceed, and that the reward is an appropriately integrable function of the distribution), and still manages to provide an algorithm that works in all such settings, with regret bounds that depend on the spectral properties of the design operator corresponding to each concrete instantiation of the setting at hand.

**Theoretical Claims:**

Yes, I checked for correctness of the overall methodology and the broad-level correctness of the analyses, and am reasonably convinced that the results are substantially true (possibly up to minor unchecked details).

---

> ### Author Rebuttal · Authors · 2025-04-01
>
> Thank you for your remarks and questions. We would now like to answer your questions in order.
>
> **RE: Squared-Integrability**
> Thank you very much for your question! We point out that in many online learning settings such as finite-dimensional linear bandits [85, Abbasi-Yadkori, Yasin et al. (2011)]
> including non-stationary bandits [74, Zhao, Peng et al. (2020)], conservative bandits [73, Kazerouni, Abbas et al. (2017)], combinatorial bandits, [84, Liu, Qingsong et al. (2022)], differential privacy bandits [83, Han, Yuxuan et al. (2021)] square-integrability is always assumed. In these problems, square-integrability is presented as finite $2$-norm assumption $\theta^*\in R^d$, $||\theta^*||_2\le S$ for some $S$.
> Our assumption is an extension because the finite $2$-norm in infinite dimensional function space is square-integrability.
> Generally speaking, assuming no prior knowledge about model class, you can still estimate using any model class.
> In this circumstance, there will be two types of errors, the first type is **approximation
> error** induced by potential model misspecification. This is unavoidable and there is no way to reduce because the assumption about the underlying model is wrong. The second type is estimation error, this can be reduced by designing more delicate algorithms. In practice, you can use deep learning and neural network with strong expressive power to approximate the underlying function to reduce approximation error. In this paper, we assume well-specified model class and focus on reducing the estimation error.
>
> **RE: $\gamma$-intuition**
> We explain the intuition of $\gamma$ from the following:
> First, the value of eigenvalue reflects the information in that direction, and larger eigenvalue means more information. Our $\gamma-$ eigendecay condition says $U_{D}=\sum_{i=1}^{\infty}\lambda_ie_i$,
> where $\lbrace\lambda_i\rbrace_{i=1}^{\infty}$ is eigenvalue sequence and we have $\sum_{i=1}^{\infty}\lambda_i^{\gamma}<s_0<\infty$.
> We design an adaptive truncation method based on the decay rate of eigenvalues to solve an infinite-dimensional problem. We achieve a utility regret rate of $\mathcal{O}(T^{\frac{3\gamma+2}{2(\gamma+2)}})$ for our algorithm. For small $\gamma$, the eigenvalue sequence $\{\lambda_i\}$ is decaying fast enough, so the information of this operator is concentrated in the first several largest eigenvalues. Therefore, our finite-dimensional eigenvalue truncation preserves most of the information stored in the original operator. So our regret $\mathcal{O}(T^{\frac{3\gamma+2}{2(\gamma+2)}})$ will be very good. If we assume no prior knowledge about the eigendecay rate instead, just use the trace-class property and set $\gamma=1$ also leads to a sublinear $O(T^{5/6})$ regret.
>
> We finally remark that under mild conditions, polynomial eigendecay for integral operator could be rigorously proved and assumption 2.10 is satisfied. See Thm 4 in [47, Carrijo, Angelina O et al. (2020)].
>
> **RE:Examples**
> Thanks for the question. We illustrate the mean as an example here. MV bandits and other applications could be derived similarly. For mean, functional $T$ is $T(F)=\int_{S}sdF(s)$. Then,
> $|T(F_1)-T(F_2)|=|\int_{S}sd(F_1(s)-F_2(s))|=|\int_{S}F_1(s)-F_2(s)ds|,$
> Then by algebra,
>
> $
> |\int_{S}F_1(s)-F_2(s)ds|\le \int_{S}|F_1(s)-F_2(s)|ds\le m(S)^{1/2}(\int_{S}|F_1(s)-F_2(s)|^2ds)^{1/2}=m(S)^{1/2}||F_1-F_2||_{L^2(S)}.
> $
> This indicates that the decision-mapping is Lipschitz continuous with respect to our $L_2$ metric.
>
> **RE: Notation** We use $F(x,a,s)$ to denote a distribution to show that we want to emphasize that the distribution is related to context action pair $x,a$. We will polish accordingly in the camera-ready version.
>
> **RE: Compare with [54, Zhang, Qian et al. (2022)]**
> In [54, Zhang, Qian et al. (2022)], they use a regularizer which scales with the number of the sample points. The main difference is that in terms of the estimator in [54, Zhang, Qian et al. (2022)], they just used the fact that  **the sum of the eigenvalues of the operator is finite** without the eigendecay rate characterization. We describe the behavior of the eigenvalue sequence in a more meticulous way and discover that the scaling regularizer is no longer needed. Instead, we can use the decay parameter $\gamma$ to depict the regret $O(T^{\frac{3\gamma+2}{2(\gamma+2)}})$.
>
> Thank you again for your comments. We hope you find the response satisfactory. Please let us know if you have further questions.
>
> [47] Approximation tools and decay rates for eigenvalues of integral operators on a general setting
>
> [54] Functional linear regression of cumulative distribution functions
>
> [73] Conservative contextual linear bandits
>
> [74] A simple approach for non-stationary linear bandits
>
> [83] Generalized linear bandits with local differential privacy
>
> [84] Combinatorial bandits with linear constraints: Beyond knapsacks and fairness
>
> [85] Improved algorithms for linear stochastic bandits

---

> > ### Comment · Reviewer_t1kP · 2025-04-09
> >
> > Thank you for your response. The biggest improvement area by far will be making improvements to the presentation/writing of the manuscript --- as I originally mentioned in my review and as also emerged from the threads with the other reviewers. That said, I am satisfied with the answers, and will keep my score.

---

### Decision · Program_Chairs · 2025-05-01

**Decision:**

Accept (poster)

**Comment:**

This paper presents a general framework for contextual online decision-making by combining well-established techniques from statistics and the bandit literature. The proposed approach, based on infinite-dimensional functional regression and inverse gap weighting enables a unified approach to various contextual online decision-making problems. However, reviewers have raised several important concerns. Firstly, the clarity of the writing must be improved. Secondly, the practical application of the proposed method, especially the crucial aspect of selecting an appropriate basis family in real-world scenarios, may be difficult. Finally, the scope and applicability of the assumed eigendecay conditions warrant a more detailed discussion and analysis.